# Real time, in vivo measurement of neuronal and peripheral clocks in *Drosophila melanogaster*

**Peter S Johnstone[1,2], Maite Ogueta[3], Olga Akay[1,2], Inan Top[4], Sheyum Syed[5], Ralf Stanewsky[3], Deniz Top[1,2]\***

[1]Department of Biochemistry and Molecular Biology, Dalhousie University, Halifax, Canada; [2]Department of Pharmacology, Dalhousie University, Halifax, Canada; [3]Institute of Neuro- and Behavioral Biology, Westfälische Wilhelms University, Münster, Germany; [4]You.i Labs Inc, Ottawa, Canada; [5]Department of Physics, University of Miami, Miami, United States

**Abstract** Circadian clocks are highly conserved transcriptional regulators that control ~24 hr oscillations in gene expression, physiological function, and behavior. Circadian clocks exist in almost every tissue and are thought to control tissue-specific gene expression and function, synchronized by the brain clock. Many disease states are associated with loss of circadian regulation. How and when circadian clocks fail during pathogenesis remains largely unknown because it is currently difficult to monitor tissue-specific clock function in intact organisms. Here, we developed a method to directly measure the transcriptional oscillation of distinct neuronal and peripheral clocks in live, intact *Drosophila*, which we term Locally Activatable BioLuminescence, or LABL. Using this method, we observed that specific neuronal and peripheral clocks exhibit distinct transcriptional properties. Loss of the receptor for PDF, a circadian neurotransmitter critical for the function of the brain clock, disrupts circadian locomotor activity but not all tissue-specific circadian clocks. We found that, while peripheral clocks in non-neuronal tissues were less stable after the loss of PDF signaling, they continued to oscillate. We also demonstrate that distinct clocks exhibit differences in their loss of oscillatory amplitude or their change in period, depending on their anatomical location, mutation, or fly age. Our results demonstrate that LABL is an effective tool that allows rapid, affordable, and direct real-time monitoring of individual clocks in vivo.

**\*For correspondence:**
dtop@dal.ca

## Editor's evaluation

This manuscript will be of broad interest primarily to readers in the field of Chronobiology, but more in general, also Physiology. The reporter construct generated in this study provides a great tool to dissect with cell and tissue specificity the rhythmic transcriptional oscillations orchestrated by circadian clocks in vivo. Here, the authors take full advantage of such a tool showing how neuronal and peripheral clocks might be differentially regulated and possess distinct properties.

## Introduction

'Circadian rhythms' collectively refer to ~24 hr oscillations in an animal's behavior and physiological responses to daily environmental changes. These rhythms are regulated by the circadian clock, a transcription/translation negative feedback loop that controls the ~24 hr oscillations in expression of hundreds of genes in every tissue. Circadian clocks are highly evolutionarily conserved time-keeping machines, from flies to humans. In both organisms, specialized neurons that express circadian clock

**eLife digest** The daily rhythms in our lives are driven by biological mechanisms called circadian clocks. These biological clocks are protein machines found in almost every cell and organ of the body, in nearly all living things, from fungi and plants to fruit flies and humans. These clocks control 24-hour cycles of gene activity and behaviour, and are kept in-time by so-called 'master clocks' in the brain.

Ideally, scientists would be able to observe how circadian clocks work in different parts of the brain in a living animal and track changes throughout the day, as the animal performs different behaviours. However, the tools that are currently available to study circadian clocks do not allow this. To overcome this difficulty, Johnstone et al. used fruit flies to develop a new method that allows scientists to measure the oscillations of the circadian clocks in the brain in real time.

Circadian clocks are composed of proteins called 'transcription factors' that activate different genes throughout the day, producing different proteins at different times. Transcription factors control the activity of genes by binding to DNA sequences called 'promoters' and switching the genes regulated by these promoters on or off.

Knowing this, Johnstone et al. engineered fruit flies to carry the gene that codes for a protein called luciferase, which emits light, and placed it under the control of the promoter for the *period* gene, a gene that is regulated by the circadian clock. To prevent all of the cells in the fly from producing luciferase any time the *period* promoter was active, Johnstone et al. placed a second gene between the promoter and the luciferase gene. This second gene contains 'stop' sequences that prevent luciferase from being produced as long as the second gene is present. Importantly, this gene can be genetically removed from specific cells in live flies, so only these cells will produce luciferase.

When Johnstone et al. removed the second gene from specific cells in the fly brain that are involved in controlling behaviours related to the circadian clocks, these cells started emitting light in cycles that reproduced the activity of the circadian clocks. Thus, by monitoring how the brightness of luciferase changed throughout the day in these flies, Johnstone et al. were able to reveal how the circadian clocks work in different parts of the fly brain.

They found that each clock had slightly different cycling lengths, suggesting that the clocks work differently in different parts of the brain to control behaviour. Interestingly, Johnstone et al. found that if a key gene responsible for communication between cells was mutated, the effects of the mutation also varied in different parts of the brain. This suggests that different clocks respond differently to communication cues. Additionally, the results showed that circadian clock activity also changed with age: older flies had weaker circadian behaviours – fewer changes in both behavioural and genetic activity levels between the day and night – than younger animals.

Johnstone et al.'s approach makes it possible to track a living animal's circadian clocks in different parts of the brain and in different organs in real time without the need to dissect the animal. In the future, this method will help scientists understand the links between different circadian clocks, the genes associated with them, and the behaviours they control.

components are considered the 'central clock'; circadian clock components in non-neuronal tissue (hereafter, 'peripheral clocks') are widely assumed to respond to the central clock (*Brown et al., 2019*; *Franco et al., 2018*; *Ito and Tomioka, 2016*; *Patke et al., 2020*; *Pilorz et al., 2018*), likely through secreted factors (*Handler and Konopka, 1979*).

In humans, disruption of the circadian clock is associated with a wide range of pathologies, including neurological, cardiovascular, and metabolic disorders, as well as cancer and aging (*Acosta-Rodríguez et al., 2021*; *Bae et al., 2019*; *Hood and Amir, 2017a*; *Hood and Amir, 2017b*; *Leng et al., 2019*; *Logan and McClung, 2019*; *Rana et al., 2020*; *Shimizu et al., 2016*; *Sulli et al., 2019*; *Thosar et al., 2018*; *Tsuchiya et al., 2020*; *Zhang et al., 2021*). Such a broad variety of pathologies associated with compromised circadian rhythms suggests a need for cheap and effective ways to measure tissue-specific circadian clocks directly in model organisms. In animals, locomotion is the simplest and most rapid way to measure circadian clock output, but this output embodies the cumulative activity of many clocks and does not necessarily represent all clocks equally. Moreover, while ablation of neuronal clocks in flies, mice, and humans leads to loss of sleep/activity rhythms and is thought to cause loss of circadian clock function in many tissues, the hierarchy of dysfunction of tissue-specific clocks during

specific disease pathogenesis remains unclear. Currently, individual peripheral clocks can be measured in flies by removing the organ and extracting RNA to assess transcriptional oscillations (*Erion et al., 2016*; *Gill et al., 2015*; *Litovchenko et al., 2021*; *Wang et al., 2004*; *Xu et al., 2011*). Such terminal qRT-PCR outputs from explanted organs measure clock function only for that time point in the lifespan of the organism and can be time-, cost-, and labor-intensive. Given that circadian clocks appear to be linked to a wide range of physiologies, including metabolism as well as various behavioral disorders, there is a need to monitor distinct cell- and tissue-specific circadian clocks directly, in vivo, and in real time in *Drosophila*, similar to reporters developed in mouse models (*Sinturel et al., 2021*; *Smith et al., 2022*).

We developed a genetically encoded reporter to monitor distinct clocks in *Drosophila* that we call <u>L</u>ocally <u>A</u>ctivatable <u>BioL</u>uminescence (LABL), offering both high spatial and temporal resolution of clock oscillations in vivo. Our data reveal that tissue-specific clocks have similar but distinct properties of oscillation. To determine if tissue-specific clocks are differentially affected by whole-body mutations, we tested flies lacking a functional PDF (pigment dispersing factor) receptor (*han*[5304], also known as *pdfr*[5304]) (*Hyun et al., 2005*) to demonstrate the properties of clock oscillations LABL can help uncover. PDF is a neuropeptide that elicits a cAMP response from most circadian neurons in the brain and is required to maintain robust rhythmic behaviour in constant dark conditions (*Helfrich-Förster, 1995*; *Helfrich-Förster et al., 2000*; *Park et al., 2000*; *Renn et al., 1999*; *Shafer et al., 2008*). While *tim*[01] flies (lacking a functional clock) become completely behaviourally arrythmic and *han*[5304] flies become mostly behaviourally arrhythmic in constant dark conditions, quantification of *han*[5304] fly neuronal clocks reveals infradian oscillations over a narrower range with a mean of ~60 hr, suggesting that loss of different circadian components can disrupt circadian locomotor activity in different ways. When peripheral clocks of a *han*[5304] mutant fly are investigated, they continue to oscillate, but with decreased stability. Here, we demonstrate that LABL reporter flies can be used to measure distinct circadian clocks in different neuronal subpopulations and peripheral tissues in real time and in vivo. The differential changes to distinct clocks caused by the *han*[5304] mutation underscores the assertion that tissue-specific circadian clocks are differentially regulated. We believe that this technology will be critical in the interrogation of distinct circadian clocks in future studies, particularly in monitoring peripheral clock function during disease progression in *Drosophila*.

## Results
### Construction of LABL
To monitor distinct clock oscillations in real time, in vivo, we designed a genetically encoded reporter that we call LABL. LABL was constructed into an attB cloning vector for *Drosophila* embryo injection

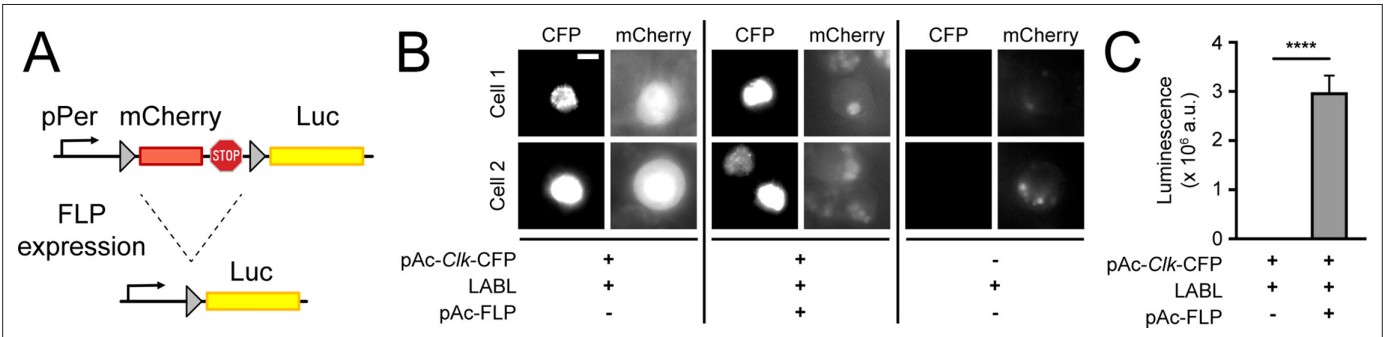

**Figure 1.** Design and activity of LABL reporter. (**A**) The Locally Activatable Bioluminescence (LABL) reporter construct. In architecture, the *per* promoter (pPer) is fused to *mCherry* followed by *luciferase* (Luc). The *mCherry* gene and three stop codons are flanked by FRT recombination sequences. Expression of Flipase (FLP) excises out *mCherry* (dashed lines), leaving luciferase under *period* promoter regulation. (**B**) Fluorescence image of LABL-expressing S2 cells. Expression of LABL reporter with actin promoter-driven *Clk* (pAc-*Clk*-CFP), FLP (pAc-FLP), or both reveals CLK-dependent expression of mCherry in the absence of FLP. Scale bar represents 5 μm. (**C**) LABL can be activated in cultured S2 cells. Lysed S2 cells emit measurable luminescence in a luciferase assay when expressing LABL reporter in a FLP-dependent manner. ****: p < 0.0001.

The online version of this article includes the following figure supplement(s) for figure 1:

**Figure supplement 1.** LABL cloning strategy.

and PhiC31-mediated genome integration (*Bischof et al., 2007*; *Figure 1—figure supplement 1*). LABL is comprised of a *per* promoter fused to mCherry flanked by FRT sequences, which was subsequently fused to Luc2 (pGL4.10) (*Figure 1A*). Three stop codons placed 3' of mCherry are designed to block Luciferase expression. The employed *per* promoter (~6.7 kb) responds to clock regulation through the CLK/CYC transcription activator complex (*Bargiello et al., 1984*; *Darlington et al., 1998*). Tissue-specific expression of Flipase (FLP) triggers recombination at the FRT sites, excising mCherry from the genome and leaving Luciferase under *per* promoter control. To test the functionality of LABL in vitro, we monitored FLP-driven LABL activity in cultured S2 cells (*Figure 1B and C*). Since S2 cells do not express CLK, which is required to activate the *per* promoter, we co-transfected LABL plasmid with *Clk*-CFP, with and without FLP. These cells were subsequently imaged for CFP and mCherry expression and monitored for luminescence. Cells expressing FLP exhibited increased luminescence and a loss of mCherry fluorescence. Cells lacking FLP exhibited no luminescence but increased mCherry fluorescence. Thus, these data demonstrate that the LABL reporter is functional as designed and can be activated by FLP expression.

## Luminescence oscillations of transcription activity reflect behavioural rhythms

Having established its functionality in cultured cells, we proceeded to assess LABL in adult flies. The LABL reporter plasmid was used to generate reporter flies which were then monitored in a luminometer using arenas designed to hold 15 flies on top of fly food supplemented with luciferin (*Figure 2A*). LABL flies carrying the *tim*-UAS-Gal4 (TUG) pan-circadian tissue driver were crossed to flies carrying UAS-FLP, and the progeny monitored for luminescence activity in constant darkness (*Figure 2B*). The raw luminescence data exhibited an oscillating rhythm and a gradual decay, as expected (*Brandes et al., 1996*; *Stanewsky et al., 1997*).

We next compared pan-circadian tissue luminescence oscillations with behavioural rhythms. The recorded luminescence activity was normalized to the gradual decay in signal and plotted as an average of four experiments (*Figure 2C*, top panel). Control flies lacking a driver exhibited no discernable oscillation (white line). To characterize the luminescence oscillations, we first quantified the decay in amplitude (the difference in y-value, between peaks and troughs) of signal. Data points were binned into 30-min time intervals and a 48-hr sinusoidal curve was fitted to the data at 24-hr intervals (*Figure 2D*). The coordinates of the local minima and maxima were recorded, averaged, and plotted over the decay-normalized luminescence signal to reveal the amplitude of oscillation across time (black circles) (*Figure 2C*). An S-curve fitted to the changing local minima and maxima (black line) revealed points of inflection coinciding with Day 6 of constant darkness (vertical dashed lines). We used this point of inflection, which is the time at which the amplitude falls to 50% between maximum and minimum (A50), as a measure of clock stability because decay of oscillations into arrhythmic transcription may exceed the timeline of this assay.

Genotypically identical flies were measured for locomotor activity and their behavioral rhythms plotted (*Figure 2C*, bottom panel). We found that the peaks of morning anticipation (yellow circles) decayed rapidly, allowing the evening anticipation peaks (blue circles) to dominate behavioral oscillations in constant dark conditions. Focusing on the change in evening anticipation peaks, we found that a fitted S-curve revealed a point of inflection (A50) at ~day 6, coincident to the A50 observed in luminescence oscillations. We conclude that the decay of amplitude of oscillation of the ubiquitously expressed molecular clock and behavioural rhythms are consistent with each other.

We next characterized the change of period of luminescence oscillation across time. A Morlet wavelet was fitted onto the measured luminescence oscillations (*Figure 2E*). Period values with highest confidence intervals revealed a steady ~23.5 hr period of oscillation across time; specifically, oscillations occurred with an average period of 23.57 hr over nine days in constant darkness. This oscillation period also mirrored the behavioural period, with luminescence oscillation and behavioural period statistically the same, at ~23.5 hr. Thus, luminescence could be detected in flies in which LABL was activated using UAS-FLP and the pan-circadian Gal4 driver TUG and revealed parallels between transcription oscillations and behavioral rhythms of the fly.

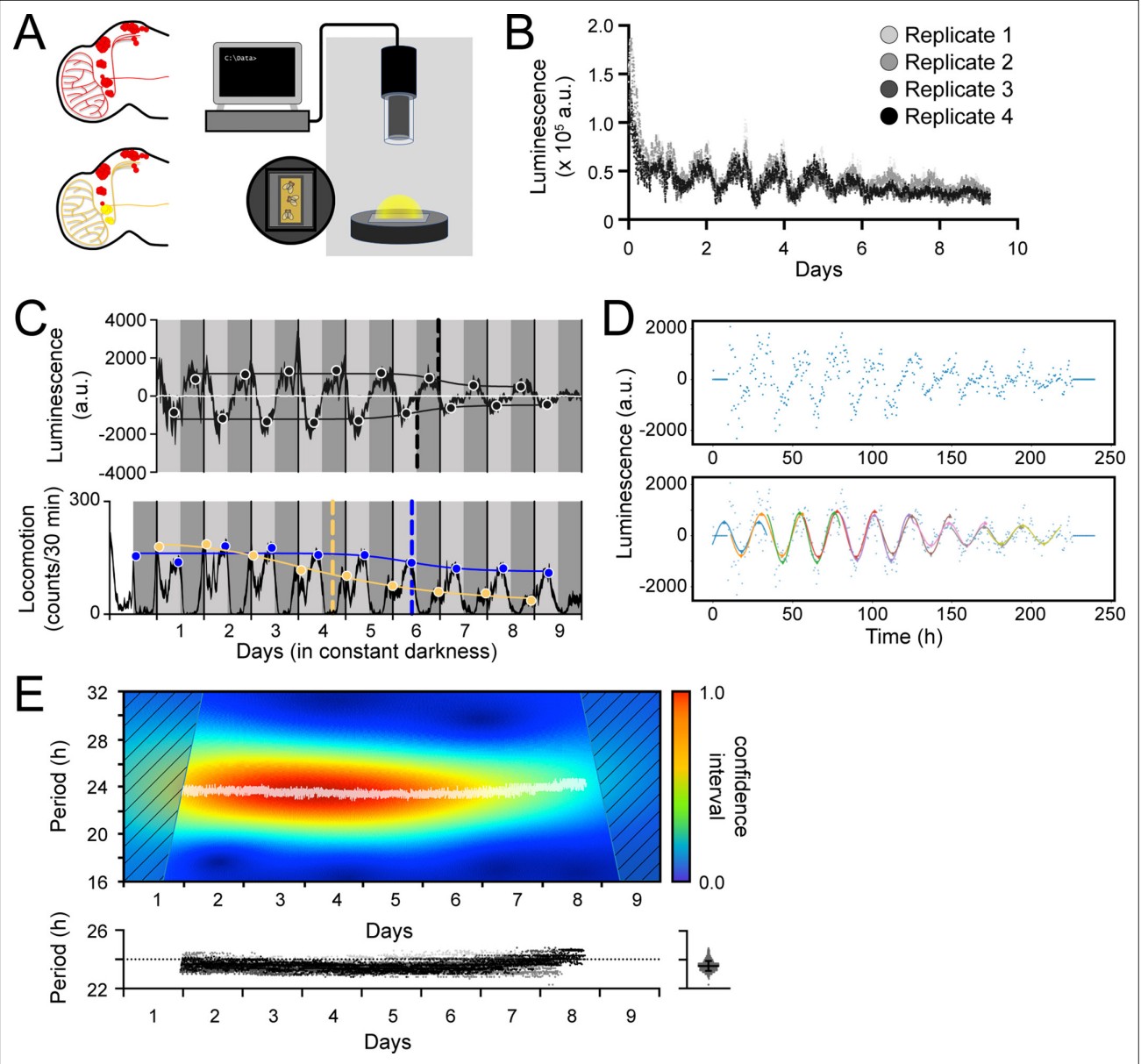

**Figure 2.** Measurement, quantification, and analysis of luminescence from LABL flies. (**A**) LABL activation strategy in *Drosophila* brain. Fly brain schematic illustrates how LABL can be activated using tissue-specific Gal4 drivers which express UAS-FLP2 to excise *mCherry* out of the genome in some neurons to permit Luciferase expression (yellow circles) under the regulation of the *period* promoter, leaving other neurons untouched (red circles). Fifteen LABL flies are placed in custom-made plates containing luciferin mixed with standard fly food. Plates are loaded into a luminometer and luminescence from each cohort is recorded for analysis. (**B**) Raw luminescence measurements of live flies. Photons were detected from four replicates of 15 flies expressing LABL reporter, UAS-FLP2 and *tim*-UAS-Gal4 over 9 days at 4-min temporal resolution. (**C**) Comparison between luminescence signal (upper graph) and locomotor activity (lower graph) of wild-type flies. Luminescence signal from flies described in panel B is normalized to exponential decay of signal, the values are averaged into 30 min bins, and the mean of the four experiments presented, +/-SEM (thickness of curve). Light grey and dark grey backgrounds represent subjective day and subjective night, respectively. Vertical solid black lines divide days. The peaks and troughs of the presented curves are represented by dots, and are the mean of four experiments, +/-SEM. Lines connecting the dots of peaks or troughs are best-fit S-curves. The vertical dashed lines define the point of inflection of each S-curve, which is used to define the A50. Locomotion activity is measured in beam breaks (counts) per 30-min bouts. Curve in lower graph represents rhythmic locomotor activity of 25 flies, +/-SEM (thickness of curve). White background indicates lights on. Vertical solid lines divide days. Light grey and dark grey backgrounds represent subjective day and subjective night, respectively. Blue and yellow dots represent peaks of "morning anticipatory" and "evening anticipatory" locomotion activity, respectively. The vertical dashed lines define the point of inflection of each S-curve, fitted to peaks of activity, which is used to define the decay of amplitude of peaks of behavior. (**D**) Calculation of oscillation peaks and troughs. A representative single replicate from experiment in panels B and C is plotted. Dots represent averaged luminescence signal in 30-min bins. A sinusoidal curve spanning 2 days is fitted to the data in 1 day increments (distinct colored curves). The peaks and

*Figure 2 continued on next page*

*Figure 2 continued*

troughs of each curve are calculated (triangles), averaged for both x- and y-values and recorded. This process is repeated for all four replicates and the resulting average is reported as shown in panel C. (**E**) Changes in oscillation period over time determined by Morlet wavelet fitting. Wavelets of different periods were fitted to luminescence signal from a single representative replicate from experiment in panel B and C, and assigned a confidence interval, across time (upper graph). Periods with highest confidence intervals at a time point (i.e. across the x-axis) were plotted as white dots. Confidence intervals of 25% or less were omitted. These values were replotted along with the other replicates (below), with varying shades of grey representing each of the four experiments. The dotted horizontal line denotes 24 hr as a point of reference. Right panel: All data points without the time dimension are plotted. Bar represents the mean, +/-SD.

The online version of this article includes the following figure supplement(s) for figure 2:

**Figure supplement 1.** Dip in luminescence signal in the first peak of transcription oscillation.

## LABL signal is comparable to ubiquitously expressed luciferase reporters and is clock dependent

To determine the effectiveness of LABL as a luminescence-based reporter, we compared TUG-activated LABL oscillations to other, ubiquitously expressed luciferase reporters. To this end, we compared our data to data collected from *plo* (*per* promoter fused to luciferase) (*Brandes et al., 1996*) and PER-BG::Luc (*per* promoter and ~2/3rds of the *per* gene fused to luciferase) (*Stanewsky et al., 1997*) flies, two other luciferase-based reporter systems (*Figure 3A*). Since TUG drives Gal4 expression in all clock cells, we expected TUG-activated LABL luminescence signal to be comparable to *plo* luminescence signal. Indeed, the oscillation of luminescence signal was comparable in amplitude, period, and phase (*Figure 3A*). The PER-BG::Luc construct, on the other hand, retains a significant portion of the PER coding region, and since there is a phase delay between *per* transcription and *per* mRNA accumulation (*So and Rosbash, 1997*; *Stanewsky et al., 1997*), we expected a phase difference between TUG-activated LABL luminescence signal and PER-BG::Luc signal. As anticipated, PER-BG::Luc luminescence signal was phase-delayed compared to both TUG-activated LABL and *plo* flies (*Stanewsky et al., 1997*). Thus, LABL is predictably comparable to both *plo* and PER-BG::Luc flies.

Eliminating the clock causes behavioural arrhythmicity and loss of transcriptional oscillation. To ensure that the transcription oscillations we observed using LABL were clock-dependent, we monitored TUG-activated LABL luminescence oscillations in a genetic background lacking functional TIM expression (*tim$^{01}$*) (*Figure 3B*). As expected, both locomotor activity and transcription oscillation of *tim$^{01}$* flies were arrhythmic (top and middle panel). Importantly, our attempt to fit Morlet wavelets to the luminescence data revealed periods ranging broadly from 48 hr to 72 hr, consistent with circadian arrhythmicity (bottom panel). Elimination of PDF signaling using *han$^{5304}$* mutant flies (expressing PDF receptor lacking its transmembrane and cytoplasmic tail) permits rhythmic behavior in the first 1–2 days of constant darkness, but ultimately results in arrhythmic behavior in a majority of flies (*Hyun et al., 2005*). When we characterized LABL oscillations in a *han$^{5304}$* background (*Figure 3C*), we found that the majority of flies became arrhythmic in their locomotor activity in the second day of constant darkness, as expected. While there was also a rapid decay in transcription oscillation, measured by LABL, closer visual inspection revealed that initial (2–3 days) ~24 hr oscillations decayed into a reproducible 60-hr infradian oscillations with a reliably narrower range of period-fit. Such a decay from an ~24 hr oscillation and coherent (higher confidence wavelet fit) 60-hr infradian oscillation of luminescence was not observable in flies lacking a functional clock (i.e. *tim$^{01}$* flies). A Kolmogorov-Smirnov test reveals that the period distribution within the 48–72 hr range of the *han$^{5304}$* mutant flies was significantly different than that of *tim$^{01}$* flies (*Figure 3B and C*, inset).

Because of the unusual observation of infradian rhythms in luminescence, we wanted to determine if we could observe these oscillations using other methods, namely by locomotion assay or immunoblotting. We first measured locomotion rhythms of wild-type (iso31), *han$^{5304}$*, and *tim$^{01}$* flies, then conducted a Lomb-Scargle analysis for behavioural period determination (*Figure 3—figure supplement 1A*). In the 16– 32 hr range, wild-type flies yielded robust ~24 hr rhythms with high power of fit, *han$^{5304}$* flies yielded behavioural periods that clustered around 24 hr with a wide distribution and a low power of fit, and *tim$^{01}$* flies yielded no discernable pattern of behavioural period and a low power of fit. We repeated the analysis for the 48–72 hr range and found two clusters of behavioural periods at approximately 60–62 hr and 50 hr for both wild type and *han$^{5304}$* flies. By contrast, *tim$^{01}$* flies showed no identifiable clustering. All flies had an equivalently low power of fit. Although tempting to suggest the 50 hr and 60 hr clustering patterns we observed in wild-type and *han$^{5304}$* flies could be considered

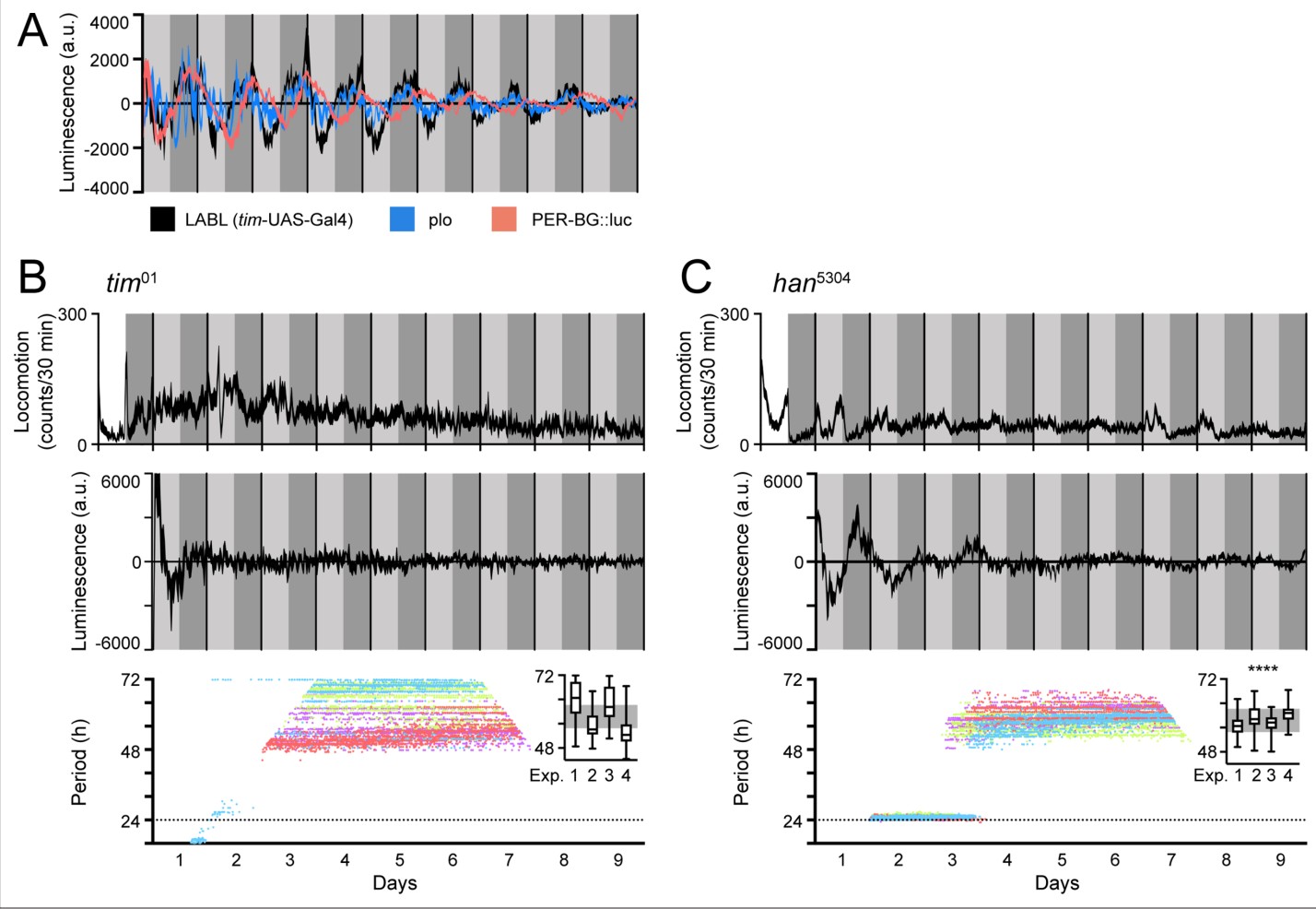

**Figure 3.** LABL oscillations are comparable to other luminescence reporters and respond to circadian clock manipulation. (**A**) Comparison of luminescence reporters. LABL activated by *tim*-UAS-Gal4 (black curve) is plotted alongside *per* promoter fused to luciferase (*plo*; blue curve) and PER-BG::Luc (pink curve). (**B**) LABL is activated by *tim*-UAS-Gal4 in a *tim*$^{01}$ genetic background. Top graph represents locomotor activity. Locomotion activity is measured in beam breaks (counts) per 30-min bouts. Curve represents rhythmic locomotor activity of 25 flies, +/-SEM (thickness of curve). White background indicates lights on. Vertical solid lines divide days. Light grey and dark grey backgrounds represent subjective day and subjective night, respectively. Middle graph illustrates luminescence signal over time. Luminescence signal is normalized to exponential decay of signal, values averaged into 30-min bins, as the mean of four experiments (n=15 flies) presented, +/-SEM (thickness of curve). Light grey and dark grey backgrounds represent subjective day and subjective night, respectively. Vertical solid black lines divide days. Lower graph illustrates changes in oscillation period over time determined by Morlet wavelet fitting. Parameters are as described in *Figure 2E*. Each dot color represents an experimental replicate of wavelet period best-fits in the *tim*$^{01}$ genetic background. Inset is a box plot of period best-fits of each of the four experimental replicates, plotted. Experimental replicates (Exp.) are numerically labelled. Grey background represents interquartile range of box plots of the *han*$^{5304}$ genetic background. (**C**) LABL is activated by *tim*-UAS-Gal4 in a *han*$^{5304}$ genetic background. Graphs are as described in panel B. Distribution of period fit in the infradian oscillation range (48 h – 72 hr) is significantly different in *han*$^{5304}$ flies when compared to *tim*$^{01}$ flies, determined by the Kolmogorov-Smirnov test (****: p < 0.0001).

The online version of this article includes the following figure supplement(s) for figure 3:

**Figure supplement 1.** Behavioural period analysis of wild-type, *han*$^{5304}$ and *tim*$^{01}$ flies.

infradian rhythms, due to the low power of fit that was comparable to that of *tim*$^{01}$ flies, we do not consider these data reliable representations of infradian oscillations.

We next extracted protein from isolated *han*$^{5304}$ fly heads, then probed for PER and TIM by immunoblotting to determine if we could observe infradian oscillations (*Figure 3—figure supplement 1B*). We used protein extracts from heads because protein extracts from whole flies did not yield oscillating protein signal by immunoblot (*Figure 3—figure supplement 1C*), possibly due to non-oscillating PER protein in ovaries in flies (*Hardin, 1994*). We observed reliable PER and TIM oscillations from head extracts from both wild-type and the first day of *han*$^{5304}$ flies in constant darkness. However, the PER

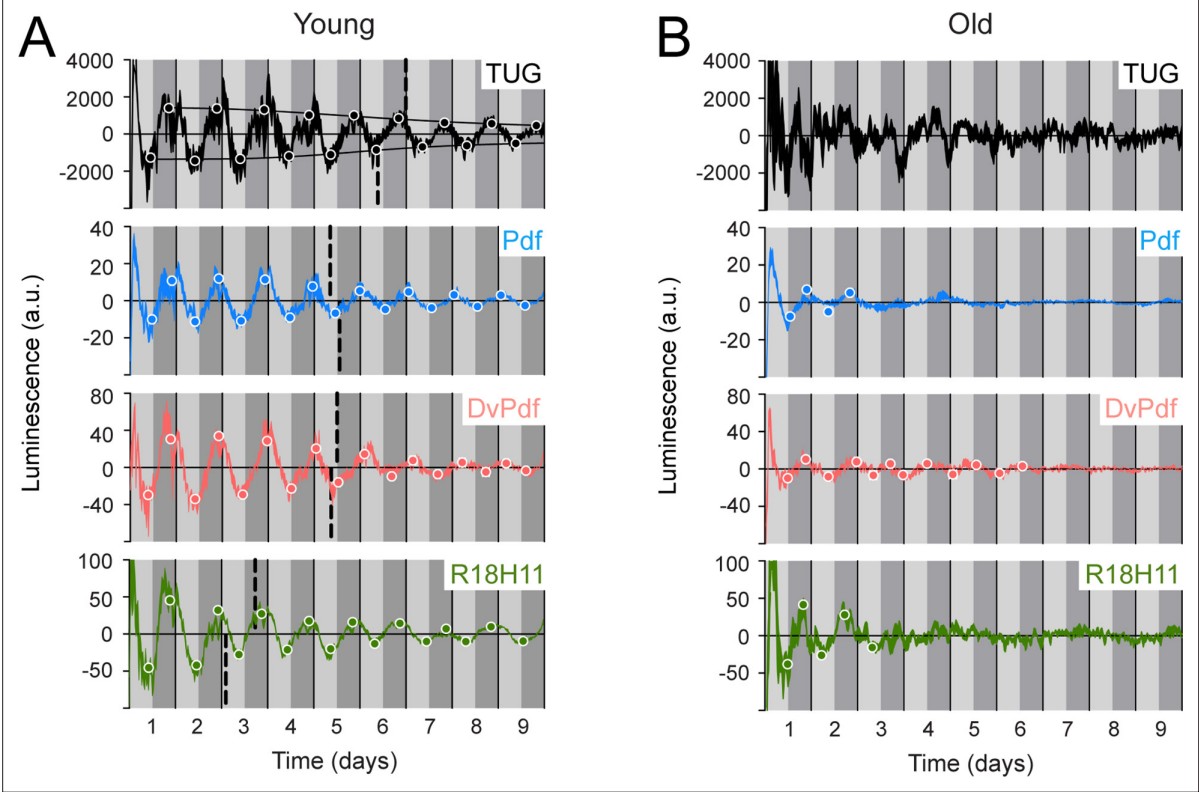

**Figure 4.** Luminescence oscillations of distinct clocks measured in young and aged flies. Luminescence signal using different Gal4 drivers to activate LABL compared in young (1–3 day old) and old (39–41 day old) flies. Drivers used to activate LABL are TUG (black), Pdf-Gal4 (blue), DvPdf-Gal4 (pink), or R18H11-Gal4 (green) in otherwise wild-type flies. Luminescence values are normalized and averaged into 30 minute bins, and the mean of four experiments presented, +/-SEM (thickness of curve). Light grey and dark grey backgrounds represent subjective day and subjective night, respectively. Vertical solid black lines divide days. The peaks and troughs of the presented curves are represented by dots, which are the mean of four experiments, +/-SEM. Where dots are absent, no peak/trough could be fitted. Lines connecting peaks or troughs are best-fit S-curves. The vertical dashed lines define the point of inflection of each S-curve, which is used to define the A50. Where a vertical dashed line is absent, either no S-curve could be fitted, or no decay could be detected.

The online version of this article includes the following figure supplement(s) for figure 4:

**Figure supplement 1.** Body drivers create ~24 hr oscillation patterns with different amplitude decay in wild-type LABL flies.

**Figure supplement 2.** Aged flies exhibit weak locomotor rhythms.

and TIM band intensity profiles did not represent reliable oscillations equivalent to a 60 hr period in these immunoblots. We conclude that the subtle amplitude of luminescence oscillation may be too small to observe by either locomotion or immunoblotting assays.

## Circadian drivers used to activate LABL reveal oscillating luminescence

Since the TUG driver can be used to activate LABL in otherwise wild type flies to record oscillating luminescence signal, we wanted to demonstrate that other commonly used circadian drivers can also be used as effectively. We drove FLP expression using Pdf-, DvPdf-, R18H11-, Clk4.1-, Mai179-, and Clk9M-Gal4 (*Figure 4A* and *Figure 4—figure supplement 1*). We found that the oscillation with the lowest signal came from flies in which LABL was activated by Pdf-, DvPdf-, and R18H11-Gal4, and this correlates well with the relatively small numbers of neurons that were targeted for LABL activation when compared with TUG-driven LABL (*Figure 4A*). Clk4.1-, Mai179-, and Clk9M-Gal4 drivers yielded oscillations with higher relative amplitudes than would be expected in proportion to their target neurons (*Figure 4—figure supplement 1*), suggesting that unintended neurons or tissues may be luminescing (see below). Oscillation periods were maintained at ~24 hr. We conclude that LABL can also be used in conjunction with more tissue-specific drivers.

## Aged flies exhibit rapid loss of amplitude of luminescence oscillation and emergence of infradian oscillations

Since luminescence oscillations in *han*[5304] exhibited differences in amplitude, period and a decay into infradian rhythms, we wanted to determine whether these changes could also be observed using LABL in a different biological context. Aged flies have weaker locomotor behaviour rhythms, lower levels of *per* transcript, and lower amplitude of luminescence signal from flies expressing luciferase by *tim* or *per* promoter, compared to their younger (1–3 day old) counterparts (*Koh et al., 2006*; *Luo et al., 2012*; *Rakshit et al., 2012*; *Umezaki et al., 2012*; *Zhao et al., 2019*, *Figure 4—figure supplement 2*). We therefore reasoned that aged LABL flies in which clocks are no longer robust should display luminescence oscillations with weak or unstable amplitude, and an altered A50, using different drivers. We aged LABL-expressing flies to 39–41 days, measured their luminescence, and compared these values to young flies (*Figure 4*). In TUG-activated aged LABL flies, we fitted appropriate peaks and troughs to the luminescence signal to determine the A50. We note that in young flies the A50 was at 6.5–7 days, almost identical to our earlier observation (*Figure 2C*), reflecting the reproducibility of this metric. However, we note that since we do not determine A50 for individual flies, this measure does not readily permit statistical analysis for comparison across genotypes. TUG-activated old LABL flies were too irregular to reliably fit peaks and troughs. When LABL was activated using more specific drivers (i.e. Pdf, DvPdf-, and R18H11-Gal4), the peaks and the troughs of old flies remained low throughout the measurement, and either did not reveal an S-curve of decay or revealed too few peaks and troughs through which to fit a curve. Thus, we conclude that an A50 could not be determined for less stable clock oscillations in aged flies because the amplitude in their endogenous clock oscillations had already decayed to their minimal stable levels.

We also note that the luminescence signal decayed into infradian oscillations in some target neurons in aged flies. On day 3 of constant darkness, Pdf- and R18H11-Gal4 activated LABL flies began to exhibit infradian oscillations. On the other hand, Dvpdf-Gal4 activated LABL flies exhibited circadian oscillations until day 7 of constant conditions, at which point the amplitude of oscillation became too low to draw any further conclusions. Thus, infradian oscillations can develop in older flies that exhibit circadian oscillations when younger, suggesting that infradian oscillations can manifest in flies that are genetically identical but distinct in age.

## Phase coupling simulations explain amplitude decay but not infradian oscillations

Dissociated neurons exhibit dampened clock oscillations measured by luminescence (*Sabado et al., 2017*), suggesting clock coupling may also influence our in vivo luminescence measurements. We therefore wanted to determine whether amplitude decay or ~60 hr infradian luminescence oscillations could emerge from oscillators that are uncoupled. We conducted a simple simulation in which half of the simulated oscillators were phase advanced by different magnitudes and the cumulative oscillation quantified (*Figure 5A*). Our simulations revealed that the magnitude of a phase advance of half the oscillators changes the amplitude of total observed oscillation but does not change the period. A 12 hr phase advance leads to destructive interference, eliminating the oscillation. We next simulated individual oscillators with the same 24 hr period that were allowed to drift in phase over time. These simulations revealed that permitting the oscillators to randomly assume different phase values dampens the overall oscillation amplitude, with a faster rate of phase drift causing a more rapid dampening effect, but still not altering the period of oscillation (*Figure 5B*). We noted that such phase drift leads to an S-curve decay of amplitude, as we observed in our biological measurements (*Figure 2C*). We conclude that allowing for phase-shifted variability of oscillations over time cannot explain the infradian rhythms seen in the data.

Thus, while rhythmic and clock-dependent, for the purposes of this report, we interpret these infradian oscillations to represent arrhythmia from the 24-hr circadian perspective (i.e. they oscillate with a period on the order of days). These data together demonstrated that the LABL signal is clock-dependent, PDF receptor (PDFR)-dependent, and that it reflects circadian locomotor behavior. Importantly, our method of analysis can quantify mutation-caused changes in transcription oscillations in target cells and tissues robustly and consistently.

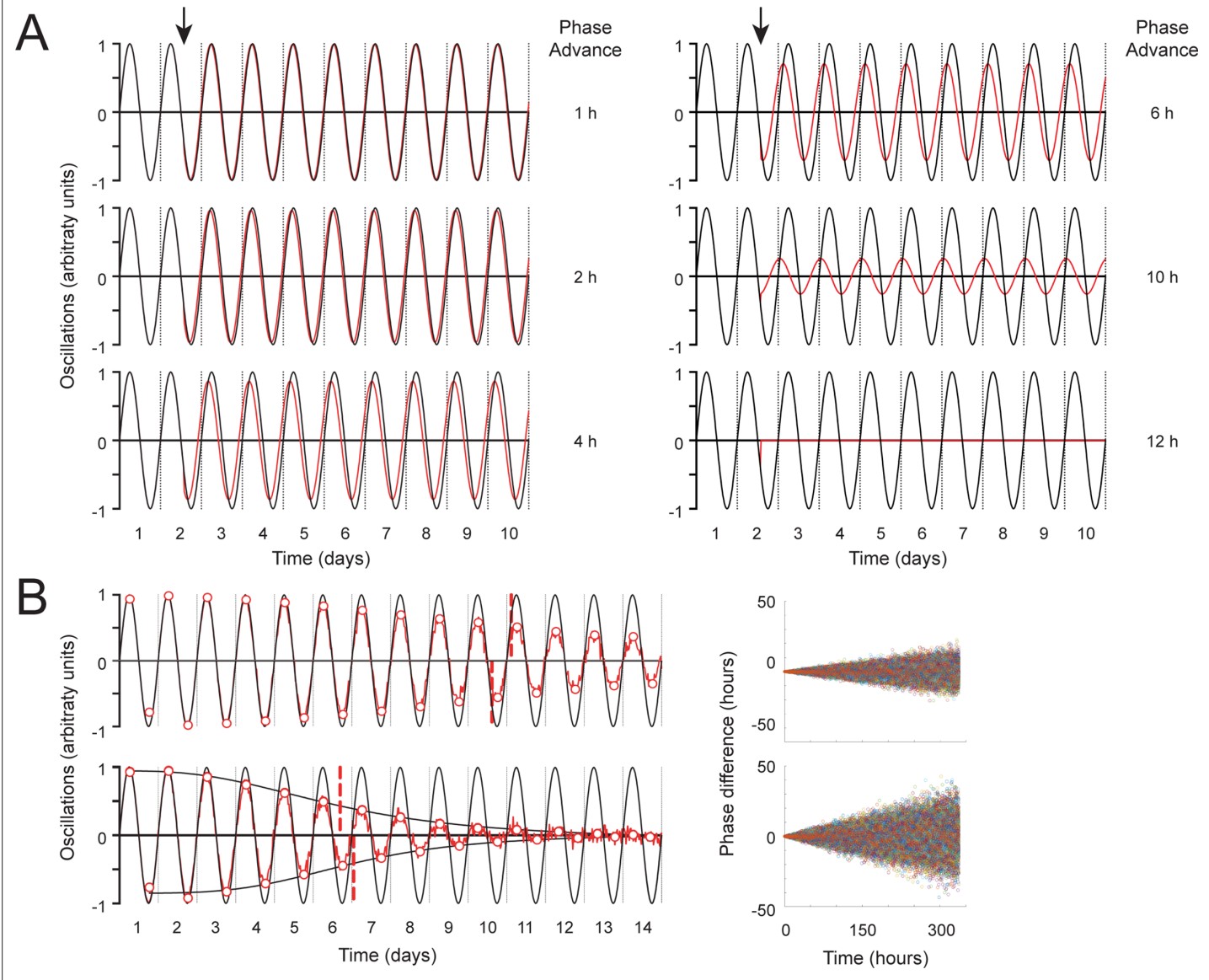

**Figure 5.** Simulations of oscillations with phase decay and measurement of distinct clocks in aged flies reveal rapidly decaying clock amplitude. (**A**) Abrupt phase change simulation. Half of 100 simulated oscillations were made to phase advance at 1.5 days (arrow) by the indicated amount of time (red line). Oscillators with no phase change (black line) are illustrated as a reference point. Y-axis represents the peak of oscillations in arbitrary units. The x-axis represents time in days. (**B**) Gradual phase change simulation. At each time point, each of 100 simulated oscillators were randomly picked from a normal distribution with mean 0 and standard deviation, σ, that increased linearly with time from σ=0 hr at t=0 hr to σ=6 hr (upper graph) or 12 hr (lower graph) at t=336 hr (red line) compared to phase-locked controls (black line). The y-axis of graphs represents the peak of oscillations in arbitrary units, while the x-axis represents time in days. White dots with red borders represent peaks and troughs of local maxima and minima. Black S-curves are fitted to these minima or maxima. The red dashed vertical line represents the point of inflection of the S-curves (A50). Graphs on the right represent the change of oscillator phases across time.

## Anatomical expression and LABL activation of some commonly used Gal4 lines

Although the anatomical locations targeted by circadian Gal4 drivers are well characterized in the brain (*Figure 6A*), the amplitude of luminescence oscillations in some of the Gal4 drivers used to activate LABL (e.g. Clk4.1-, Mai179-, and Clk9M-Gal4) suggests that these driver lines may express Gal4 in non-neuronal tissue or non-circadian neurons as well. To establish the anatomical regions in which different Gal4 lines activate LABL, we visualized luminescence in whole animals, distinguishing between Gal4 lines that visibly activate LABL in the body from those which do not.

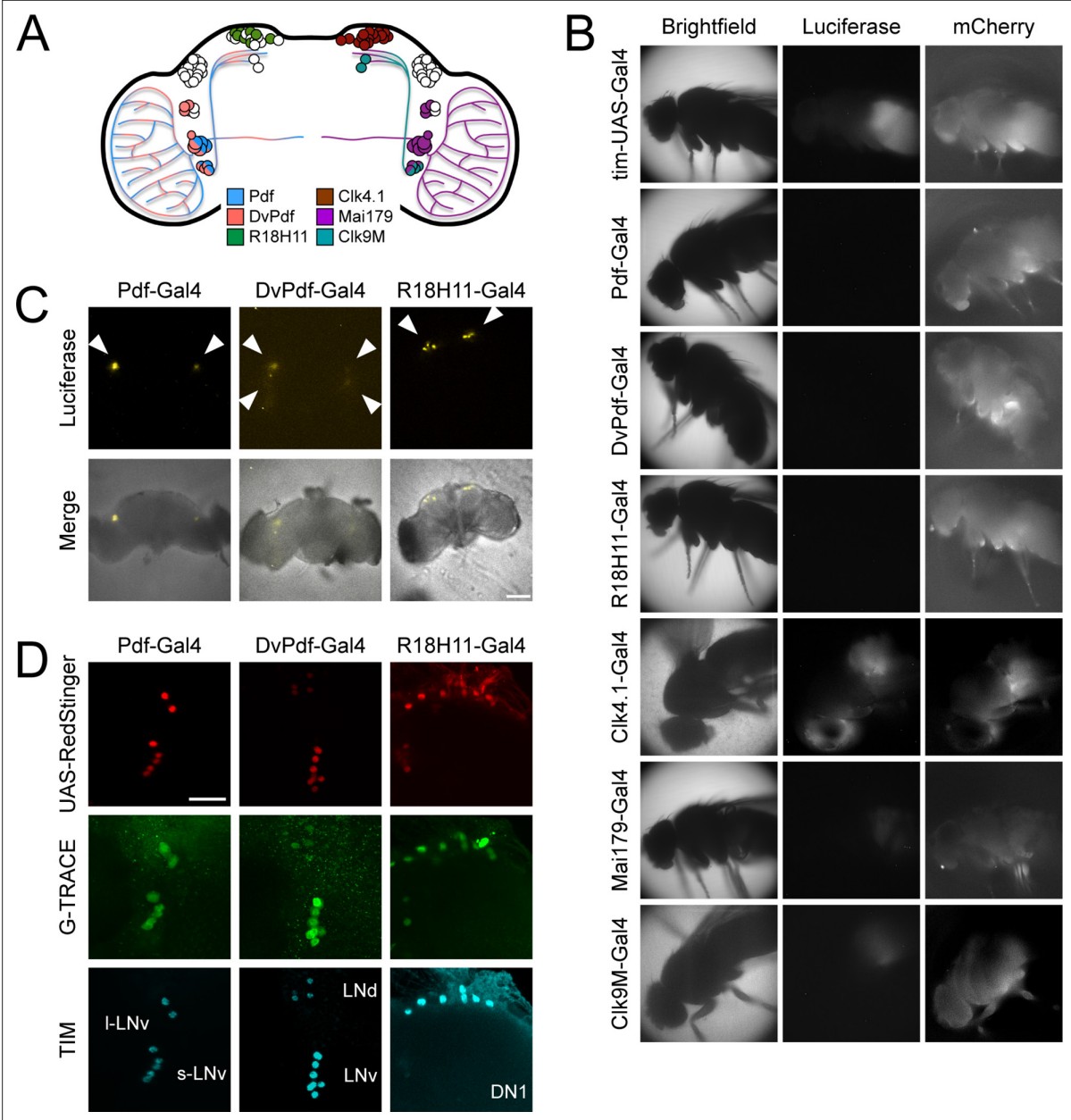

**Figure 6.** Different circadian Gal4 drivers activate LABL in distinct neurons and tissues. (**A**) Schematic of a *Drosophila* brain and Gal4 driver expression patterns. Pdf-Gal4 (blue): LNvs. DvPdf-Gal4 (pink) and Mai179-Gal4 (purple): LNvs, 3 LNds and 5th s-LNv. R18H11-Gal4 (green) and Clk4.1-Gal4 (brown): DN1s. Clk9M-Gal4 (teal): DN2s and small LNvs. (**B**) Luminescence and fluorescence images of whole flies. Flies are shown in bright field (first column), imaged for luminescence (Luciferase; second column) and fluorescence (mCherry; third column). Drivers described in panel A were used to activate LABL in the flies, as indicated. (**C**) Luminescence images of explanted brains. Explanted live brains were imaged for luminescence (Luciferase; top row) and imaged in bright field merged with luminescence (bottom row) to determine sites of LABL activation by the indicated Gal4 drivers. White arrow heads denote estimated location of clock neurons: LNvs in Pdf-Gal4 neurons, LNvs and LNds in DvPdf-Gal4 neurons and DN1s in R18H11-Gal4 neurons. Scale bar represents 50 μm. (**D**) Sites of Flipase activity in fly brains. Fly lines expressing UAS-RedStinger (red fluorescence protein, shown in red) and the indicated driver were crossed into G-TRACE flies and brains were stained for TIM (cyan), RedStinger (red) and GFP (G-TRACE; green). Identity of fluorescence neurons labelled in bottom row. l-LNv: large ventral lateral neurons. s-LNv: small ventral lateral neurons. LNd: dorsal lateral neurons. DN1: dorsal neurons 1. GFP expression is indicative of FLP activity. Pdf-Gal4 driver targets LNvs, DvPdf-Gal4 driver targets LNvs, 5th s-LNv, 3 LNds, and R18H11-Gal4 driver targets DN1s. Scale bar represents 20 μm.

Flies in which LABL was activated using Clk4.1-, Mai179-, and Clk9M-Gal4 exhibited luminescence signal from the body, similar to that of TUG (*Figure 6B*). On the other hand, the Pdf-, DvPdf-, and R18H11-Gal4 drivers showed no luminescence in peripheral tissues (*Figure 6B*). We therefore monitored luminescence from explanted live fly brains from Pdf-, DvPdf-, and R18H11-Gal4 activated LABL flies (*Figure 6C*). We found that Pdf-Gal4 activated LABL flies emitted luminescence from a single point located in the ventral lateral region of the brain, where the LNvs are expected to be. DvPdf-Gal4 activated LABL flies emitted luminescence from a wider area in the lateral region of the brain, suggesting that the sources of light are the LNvs and LNds. R18H11-Gal4 activated LABL fly brains emitted luminescence from the dorsal area, in the region DN1s are located. These observations correlated well with expected expression pattern of these Gal4 lines (*Figure 6A*). To ensure that the luminescence signal recorded from the brain specifically reflected the circadian neuronal cluster in which the Gal4 drivers were expected to activate LABL, we tested for FLP activity using G-TRACE reporter flies co-stained for TIM expression (*Figure 6D*). G-TRACE consists of the Ubip63 promoter upstream of stop codons flanked by FRT sequences, followed by the GFP gene (*Evans et al., 2009*). RedStinger (RFP)-positive neurons revealed Gal4-active sites, as shown. GFP-positive neurons that co-stain with TIM antibody revealed that the LNvs (Pdf-Gal4), the LNvs, 5th s-LNv and 3 LNd neurons (DvPdf-Gal4), and DN1s (R18H11-Gal4) had the expected FLP activity. Thus, Pdf-, DvPdf-, and R18H11-Gal4 lines activated LABL in the predicted neuronal clusters (collectively, 'brain drivers'), indicating that they are suitable for use with LABL to monitor neuron-specific clocks. The remaining drivers are collectively referred to as 'body drivers', due to their activity in the fly body.

## Circadian clock mutants $per^S$ and $per^L$ reveal features of clock oscillations in distinct neurons

The classical circadian clock mutations, *period short* ($per^S$) and *period long* ($per^L$) shorten and lengthen rhythmic behaviour, respectively (*Konopka and Benzer, 1971*). We wanted to further validate LABL using these clock mutants. The $per^S$ mutation produces a residue substitution that blocks phosphorylation of PER by DBT (*Chiu et al., 2008*; *Kivimäe et al., 2008*). This mutation in *per* leads to an early destabilization of the protein before dawn and higher amplitude of luciferase oscillation under the control of the *per* promoter, causing advanced evening anticipation and short behavioural rhythms (*Edery et al., 1994*; *Li and Rosbash, 2013*; *Marrus et al., 1996*; *Top et al., 2018*). When assessed using LABL, $per^S$ mutant flies indeed exhibited the expected short-period oscillations of luminescence (*Figure 7—figure supplement 1A–1D*). Each period of luminescence oscillation correlated well with period of behavioural rhythms. The amplitude of luminescence oscillation measured in $per^S$ flies in which LABL was activated using TUG driver was greater than their wild-type counterparts (*Figure 2* and *Figure 7—figure supplement 1A*), as expected (*Top et al., 2018*). This effect was not present in pdf-Gal4 activated and R18H11-Gal4 activated LABL flies (*Figure 4A* and *Figure 7—figure supplement 1B and D*). Amplitude stability was also differently affected across different neuronal clusters, with the $per^S$ mutation advancing the A50 by a few days in Pdf-Gal4 activated LABL flies and in DvPdf-Gal4 activated LABL flies, while R18H11-Gal4 activated LABL flies maintained a more stable amplitude over time (*Figure 4A* and *Figure 7—figure supplement 1B–1D*).

The $per^L$ mutation is a residue substitution in the PER PAS domain that interferes with TIM binding (*Gekakis et al., 1995*). As suggested by its name, this mutation causes a long behavioural period under free-running conditions (*Konopka and Benzer, 1971*). This mutation also affects temperature compensation ability in the circadian clock and leads to weaker clock oscillations (*Huang et al., 1995*; *Konopka et al., 1989*; *Rutila et al., 1996*). In LABL flies with a $per^L$ genetic background, we found that all oscillations were noisier, with lower oscillatory amplitude and longer period, consistent with these earlier reports (*Figure 7—figure supplement 1E–1H*). The apparent noisiness of the oscillation prevented us from determining an A50, which suggests that this parameter is limited to more stable oscillations. We also found that in DvPdf-Gal4 activated LABL flies, signal from luminescence oscillations became too weak on day six of constant darkness to reliably measure, suggesting that the collective signal from these neurons does not detectably oscillate. Our data confirm the features of clock oscillations in $per^S$ and $per^L$ genetic backgrounds, but also uncover differences in distinct neuronal clusters, further validating LABL as a reliable tool to measure distinct circadian clocks.

## Elimination of PDF signaling has variable effects on neuronal clocks

Elimination of the clock component *tim* causes arrhythmic locomotor activity and clock oscillations as measured by LABL (*Figure 3B*). Elimination of PDF signaling leads to arrhythmic locomotor activity in the majority of flies and a transition from ~24 hr to narrow-range infradian oscillations (*Figure 3C*). This loss of PDF signaling also causes loss of clock synchrony between clock neurons within the fly brain (*Lin et al., 2004*; *Yoshii et al., 2009*). We therefore sought to directly measure clock activity in distinct circadian neurons through activation of LABL in mutants lacking PDF signaling (*han⁵³⁰⁴*). Pdf- and DvPdf-Gal4 retained transcription oscillation in the absence of PDF signaling, with R18H11-Gal4 retaining oscillations until the end of Day 6, while Clk4.1-, Mai179-, Clk9M-Gal4, and TUG began to deteriorate after one cycle of oscillation (*Figure 7A*), in parallel with observed development of arrhythmic behaviour in locomotor output (*Figure 3C*). Thus, signal from LABL flies activated by body drivers appears to deteriorate relatively rapidly. In all cases, loss of PDF signaling in the *han⁵³⁰⁴* mutants revealed higher amplitude of luminescence oscillation, to varying degrees, relative to wild type, consistent with earlier observations of high amplitude PER-LUC oscillations in subsets of clock neurons in a *pdf⁰¹* genetic background (*Versteven et al., 2020*; *Yoshii et al., 2009*).

Decay of oscillation amplitude is an indicator of clock robustness. We therefore quantified the A50 value for the different neurons and tissues in which LABL was activated. Stability of luminescence oscillation is represented as a bar (see top of luminescence recordings in *Figure 7A*), with the dark region representing the early minima/maxima within the S-curves, the light region representing the later minima/maxima within the S-curve, and the gradient representing the difference between points of inflection measured. A comparison of A50 values of different oscillators revealed that it can range from 7 days, in the case of Mai179-Gal4 activated wild-type LABL flies, to 2 days in Clk9M-Gal4 activated wild-type LABL flies (*Figure 7B* and *Figure 4—figure supplement 1*). Although we have not determined the extent of influence of the genetic background the GAL4 drivers have on the A50, in all cases elimination of functional PDFR shifted the A50 earlier in constant dark conditions, or eliminated it altogether. *han⁵³⁰⁴* flies with Pdf- and DvPdf-Gal4 activated LABL maintained their oscillations despite an advanced A50 (*Figure 7A and B*), in agreement with earlier immunofluorescence data that demonstrate LNvs and LNds continuing to oscillate in the absence of PDF signaling (*Lin et al., 2004*). *han⁵³⁰⁴* flies with Pdf-Gal4 activated LABL exhibited a high amplitude relative to their wild-type counterparts for ~5 days. This is similar to an observed increase in *tim* promoter activity in s-LNvs in flies lacking PDF signaling (*Mezan et al., 2016*). On the other hand, DvPdf- and R18H11-Gal4 activated *han⁵³⁰⁴* LABL flies exhibited amplitude peaks similar to their wild-type counterparts after the first cycle of oscillation. Interestingly, with loss of functional PDFR, R18H11-Gal4 activated LABL flies had a relatively unchanged (~half day difference) A50 but relied on PDF signaling to maintain rhythmicity (becoming arrhythmic after Day 6). The large A50 advance observed in DvPDF-Gal4 activated *han⁵³⁰⁴* LABL flies is possibly due to phase dispersal in the s-LNvs and phase advance in the LNds caused by a loss of PDF signaling (*Lin et al., 2004*). Overall, our data suggest that body-driver activated LABL flies are dependent on functional PDFR to maintain rhythmic oscillations. Of the brain drivers, R18H11-Gal4 activated LABL flies were dependent on functional PDFR to maintain rhythmic oscillations in the long term (beyond 6 days), but Pdf- and DvPdf-Gal4 activated LABL flies continued to oscillate in flies lacking PDF signaling, as also observed in *pdf⁰¹* flies (*Yoshii et al., 2009*). From these data, we confirm that PDF signaling is critical for rhythmic locomotor activity and some neuronal clocks, but is not necessary to sustain the 'master clock' neurons (LNvs) (*Lin et al., 2004*; *Yoshii et al., 2009*).

Flies lacking PDF (*pdf⁰¹*) exhibit shorter behavioural period as well as advanced PER nuclear translocation in LNds (*Lin et al., 2004*; *Renn et al., 1999*). We therefore quantified changes in period of transcription oscillations in the different clocks to determine if shortened behavioural rhythms were caused by shortened oscillatory periods in distinct clocks. Examination of the averaged peaks and troughs of transcription oscillation revealed a consistent phase advance in PDF- and R18H11-Gal4 activated LABL flies lacking functional PDFR (*Figure 7A*). Surprisingly, elimination of functional PDFR caused longer observed clock oscillation periods in different regions of the brain, except those where R18H11-Gal4 was used to activate LABL flies, which maintained wild-type-like oscillatory periods (*Figure 7C*). This is inconsistent with earlier reports using 8.0-luc which shows a shorter oscillatory period (*Versteven et al., 2020*) and the short behavioural period of *pdf⁰¹* and *pdfr* mutants (*Hyun et al., 2005*; *Renn et al., 1999*). However, this reporter lacks the *period* promoter (but encodes the

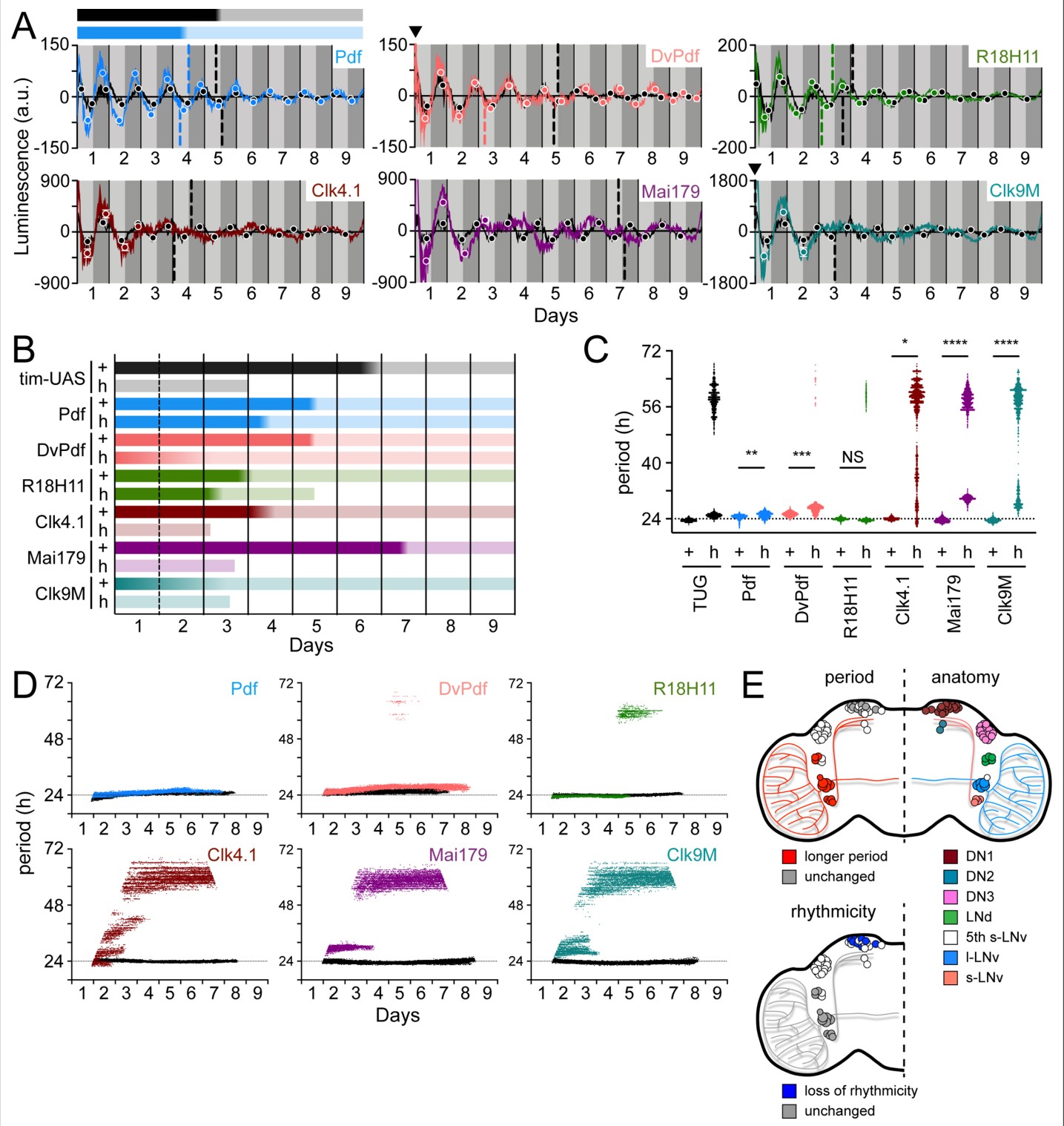

**Figure 7.** Different circadian Gal4 drivers create distinct oscillation patterns that respond to loss of PDF signaling differently. (**A**) Luminescence oscillations measured from LABL flies activated by the indicated Gal4 drivers, in a wild-type or PDF signaling deficient genetic background, plotted over time. Colored curves represent signal from flies with no PDF signaling (*han5304*) compared to their wild-type counterparts (black curves). Drivers used are Pdf-Gal4 (light blue), DvPdf-Gal4 (pink), R18H11-Gal4 (green), Clk4.1-Gal4 (brown), Mai179-Gal4 (purple), and Clk9M-Gal4 (teal). Otherwise, graphs are as described in *Figure 2C*. Bar above the curve is a visual representation of the amplitude decay across time: color fade represents the A50 calculated from the peaks and troughs. Black bar represents amplitude decay of Pdf-Gal4 activated oscillations in wild-type flies; blue bar represents amplitude decay of Pdf-Gal4 activated oscillations in *han5304* mutant flies. Location of A50 with a value close to x=0 that are not readily visible are

*Figure 7 continued on next page*

*Figure 7 continued*

indicated with a black triangle. Where A50 values are absent, no reliable S-curve fit was made (i.e. too few data points). (**B**) Stability of luminescence oscillations from LABL flies expressing different drivers are quantified in a wild-type (+) and PDF signaling-deficient (*han⁵³⁰⁴*; h) background. Colors correlate with Gal4 drivers described in panel A. Color fade in the bars represent the A50 calculated from the peaks and troughs. Faded color through entirety of bar illustrates no A50 calculation due to too few data points for S-curve fit. Length of bar indicates time at which luminescence oscillation becomes arrhythmic. (**C**) Differences in average period of luminescence oscillation measured by Morlet wavelet fitting, comparing wild-type (+) and PDF signaling deficient (*han⁵³⁰⁴*; h) flies. Colors correlate with Gal4 drivers described in panel A. If arrhythmic (>48 hr) period fits are excluded, periods of oscillations measured in the brain for wild-type and *han⁵³⁰⁴* flies can be averaged, +/-SD. Wild type: Pdf, 24.53 +/-0.61; DvPdf, 25.33 +/-0.58; R18H11, 23.93 +/-0.33 hr. *han⁵³⁰⁴*: Pdf, 25.08 +/-0.78; DvPdf, 26.79 +/-1.02; R18H11, 23.58 +/-0.30 hr. The dotted horizontal line denotes 24 hr as a point of reference. Periods of oscillations between *han⁵³⁰⁴* and wild-type flies before decay into arrhythmic or infradian oscillations (i.e., <48 hr) are averaged (black bars), and statistical significance determined between four experimental replicates (t-test). *: p<0.05, **: p<0.01, ***: p<0.001, ****: p<0.0001, NS: not significant. (**D**) Changes in oscillation period over time determined by Morlet wavelet fitting. Colored dots represent best fits from flies with no PDF signalilng (*han⁵³⁰⁴*) compared to their wild-type counterparts (black curves). Colors correlate with Gal4 drivers described in panel A. The dotted horizontal line denotes 24 hr as a point of reference. (**E**) Schematic summary of changes in period and rhythmicity of luminescence oscillation in distinct parts of the fly brain caused by loss of PDF signaling (*han⁵³⁰⁴* genetic background). LNvs and 3 LNds exhibit longer luminescence oscillation period (top left hemisphere, red) but remain rhythmic (bottom left hemisphere, grey). Luminescence oscillation in DN1s maintain the same period with loss of PDF signaling (top left hemisphere, grey), but become arrhythmic over time (bottom left hemisphere, blue). Anatomical location of s-LNvs (small ventral lateral neurons), l-LNvs (large ventral lateral neurons), 5th s-LNv (5th small ventral lateral neuron), LNds (dorsal lateral neurons), and DN1s, DN2s, DN3s (dorsal neurons 1, 2, 3), as shown (top right hemisphere).

The online version of this article includes the following figure supplement(s) for figure 7:

**Figure supplement 1.** Circadian clock mutant flies *per^S* and *per^L* reveal distinct oscillatory properties in different parts of the brain.

entire PER protein) and is expressed only in some circadian neurons (*Veleri et al., 2003*), offering a possible explanation for this discrepancy. Our data are also inconsistent with previously published locomotor data that show a shortened period of *pdf⁰¹* flies, when arrhythmic animals are excluded (*Renn et al., 1999*). Therefore, the observed shorter behavioural rhythms may be caused by (1) other clocks that oscillate with a shorter period, (2) a phase change of some of the clocks that normally converge to regulate behavioural rhythms, uncoupled by the loss of PDF (e.g., *Figure 5*), or (3) elimination of PDFR not entirely reflecting the effect of a *pdf⁰¹* fly. Indeed, flies lacking a functional PDFR (*han⁵³⁰⁴*) or PDF (*pdf⁰¹*) exhibit a phase delay or advance, respectively, of *tim* or *per* transcription oscillation in oenocytes, and PDFR is responsive to other neuropeptides (e.g. DH31), which may explain the differences that we have observed (*Krupp et al., 2013*; *Mertens et al., 2005*).

Periods of luminescence oscillation of different tissues in *han⁵³⁰⁴* flies were lengthened to different degrees. Luminescence from Mai179-Gal4 activated LABL in *han⁵³⁰⁴* flies exhibited a~30 hr period before becoming arrhythmic on day 3, as compared to a~24 hr period in wild-type flies (*Figure 7C*). A time course revealed that all oscillators that lose rhythmicity due to loss of PDF signaling appeared to form narrow range infradian rhythms (*Figure 7D*). Of these, the body drivers (Clk4.1-, Mai179-, Clk9M-Gal4) appeared to show a more disorganized period decay of period in constant darkness (*Figure 7D*). Importantly, we conclude from these data that LABL can demonstrate that PDF signaling has distinct effects on different neurons and tissues in clock stability, period, and amplitude (*Figure 7E*).

## Peripheral clocks exhibit distinct oscillations and partial PDF-dependence

In mice with suprachiasmatic nucleus (SCN; mammalian master clock neurons) lesions, organs become asynchronous (*Sinturel et al., 2021*). In flies, the PDF-expressing ventral lateral neurons are considered to be the master clock neurons (*Grima et al., 2004*; *Stoleru et al., 2004*). We reasoned that eliminating PDF-signaling would serve to reduce the master clock influence on peripheral organs in flies. This may explain, in part, the difference we observe in LABL activated by brain clocks compared to body clocks (*Figure 7*). Although PDFR protein has not been shown to be expressed in the peripheral organs, *pdfr* mRNA has (*Li et al., 2022*), suggesting that clocks in these tissues may partially depend on PDF-signaling. To characterize tissue-specific peripheral clock drivers, we monitored the stability of tissue-specific clock function in constant dark conditions in both wild-type controls and *han⁵³⁰⁴* mutants. Specifically, we examined LABL oscillations using the neuronal driver elav-Gal4 (*Saitoe et al., 2001*; *Schuster et al., 1996*; *Sink et al., 2001*), intestinal drivers esg- and NP3084-Gal4 (*Bonnay et al., 2013*; *Croker et al., 2007*; *Lee et al., 2016*; *Strand and Micchelli, 2011*), fat body drivers C564- and LSP2-Gal4 (C564 also expresses in hemocytes and some male reproductive tissues)

(*Hrdlicka et al., 2002*; *Paredes et al., 2011*; *Takeuchi et al., 2015*; *Zaidman-Rémy et al., 2006*), and muscle driver mef2-Gal4 (also expresses in some neurons [LNvs]) (*Blanchard et al., 2010*; *Cripps et al., 1998*; *Gajewski et al., 1997*, *Figure 8A*). These drivers activated LABL to reveal wild-type clock oscillations with ~24 hr periods (*Figure 8—figure supplement 1*).

If sufficient amounts of PDFR protein for signaling efficacy are not expressed in these organs despite measured *pdfr* transcript (*Li et al., 2022*), oscillations may be expected to remain undisrupted in the *han5304* genetic background. However, our observation of high-amplitude luminescence oscillations in flies lacking PDF signaling is consistent with earlier high-amplitude observations (*Versteven et al., 2020*; *Yoshii et al., 2009*) and our own observations (*Figure 7A*) that PDF signaling is relevant in these tissues (*Figure 8B*). Furthermore, the noisiness of these oscillations in flies lacking PDF-signaling, similar to hepatocyte oscillations in SCN-lesioned mice (*Sinturel et al., 2021*), suggests that PDF signaling from the master clocks plays some type of stabilizing role for these peripheral clocks.

The measured peripheral clocks appear to respond differently to a loss of PDF signaling. We found that the fat body drivers, C564-, and LSP2-Gal4, revealed phase-advanced luminescence oscillations in *han5304* flies, but these oscillations were otherwise similar to wild-type. NP3084-Gal4 driven LABL lost rhythmic oscillations within 5 days, whether or not PDF-signaling was present. Interestingly, mef2- and esg-Gal4 activated *han5304* LABL flies exhibited a loss of measurable oscillation between days 5–6 and 4–6, respectively, followed by a re-established rhythm and also exhibited earlier A50 values compared to their wild-type counterparts (*Figure 8B and C*). We also note that oscillations in the mef2- and esg-Gal4 driven LABL flies, when rhythms are reestablished, reappear in reverse phase. This echoes an observation made in the *cry*- LNds, which normally oscillate in reverse phase with respect to the *cry*+ LNds, but oscillate in synchrony with them in a *pdf01* background (*Yoshii et al., 2009*). Together, our data suggest that PDF signaling has varying influence on different peripheral clocks.

Quantification of clock periods in peripheral tissues revealed distinct periods in wild-type flies (*Figure 8D*). Importantly, LABL activated by fat body-specific drivers (C564- and LSP2-Gal4), but also for intestine driver NP3084, exhibited oscillation periods greater than 48 hr, suggesting that these clocks may not be as robust (e.g. compared to elav+ clocks) in constant darkness. Elimination of PDF signaling appeared to destabilize the period of oscillation of these and all other measured peripheral clocks to varying degrees (*Figure 8D*). To uncover the extent of period stability in the absence of PDF signaling, we monitored changes in period of peripheral clocks over time (*Figure 8E*). Intestine-specific Gal4 drivers (esg- and NP3084-Gal4) revealed transcriptional oscillations that stabilized in infradian oscillations in flies lacking PDF signaling, similar to that observed using body clock drivers (*Figure 7D*). Of note, NP3084-Gal4 activated LABL flies also exhibited infradian oscillations in a wild type PDFR background. On the other hand, muscle- and fat body-specific drivers (mef2-, C564-, and LSP2-Gal4) appeared to maintain a rhythmic ~24 hr transcription oscillation, though these were unstable, as demonstrated by intermittent deviations (*Figure 8E*) and observed visually (*Figure 8B*). Interestingly, periods of clock oscillation in the fat body were shorter in the absence of PDF signaling as compared to the wild type (*Figure 8B and D*), contrasting the longer periods observed in the same mutant background in Pdf- and DvPdf-Gal4 activated LABL flies (*Figure 7C*), but similar to shorter behavioural periods measured by locomotion (*Hyun et al., 2005*). Thus, the LABL reporter system is able to distinguish between loss of rhythmic oscillation and destabilized oscillation, as well as changes in the period of transcription oscillation, of circadian clocks in peripheral tissue.

## Discussion

Understanding the links among circadian clocks, the hierarchical organization of circadian clocks within the circadian neuronal network, and the influence which defective clocks have on behavioral/metabolic disorders requires targeted measurement of distinct clocks. Here, we present a method in which we expressed Luciferase under the control of a circadian promoter in targeted cells or tissues, allowing us to directly quantify the oscillatory properties of distinct clocks. Using this method, we found that clocks in different tissues oscillated differently, underscoring two important features of *Drosophila* circadian clocks: (1) circadian clocks have distinct oscillatory properties; and (2) that individual clocks can respond differently to some mutations, suggesting that they are differently regulated.

Distinct neuronal populations that express circadian clocks have different roles in regulating circadian behaviour. Previous work by others analyzing locomotion patterns found that LNvs regulate morning anticipatory behaviour in LD and rhythmic locomotion in constant darkness, and (CRY+) LNds

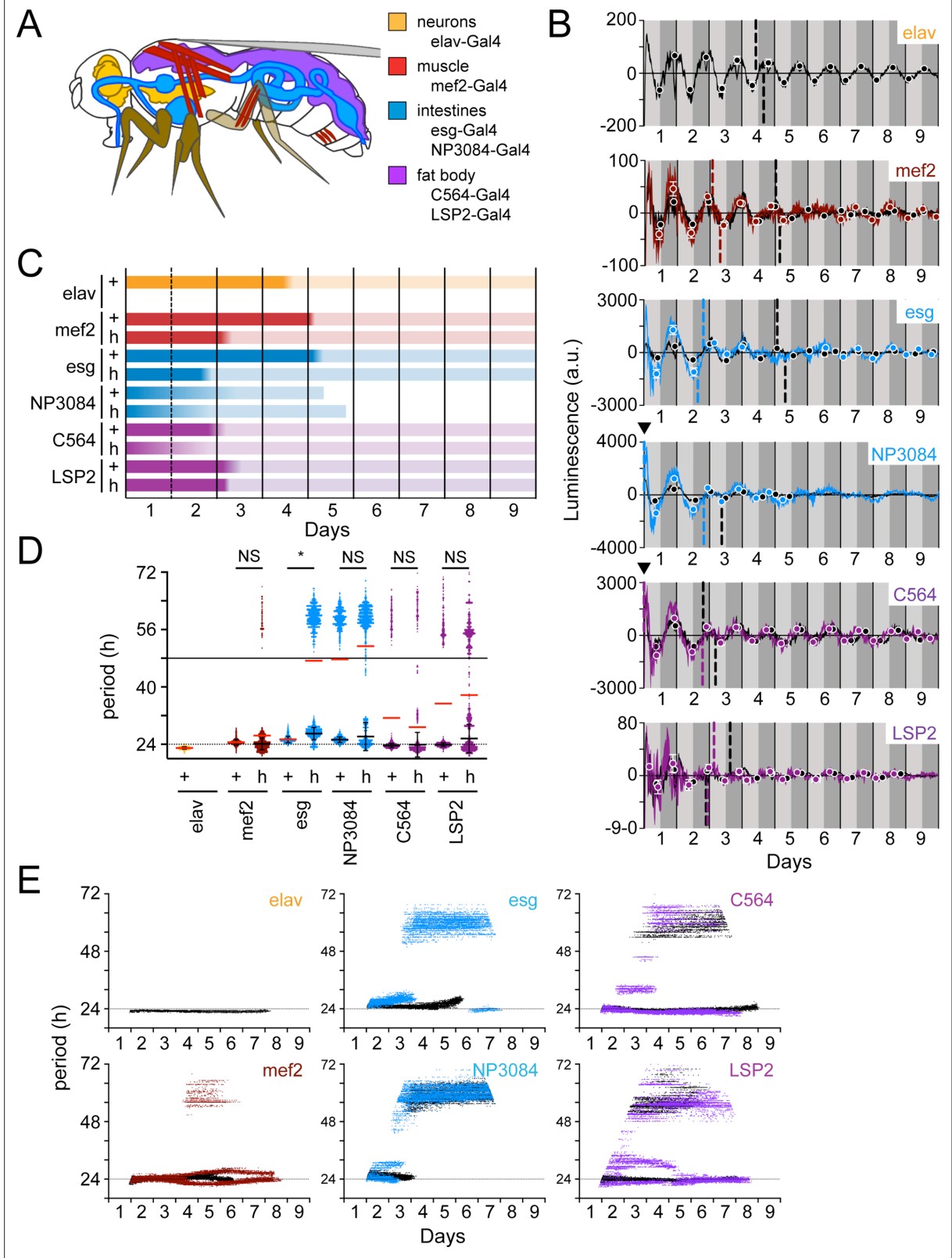

**Figure 8.** Peripheral clocks have distinct characteristics in a wild-type and PDF receptor mutant genetic background. (**A**) Schematic of target expression pattern of peripheral tissue Gal4 drivers in *Drosophila*. elav-Gal4 (neurons; orange), mef2-Gal4 (muscle; red), esg-Gal4 and NP3084-Gal4 (intestines; blue), and C564-Gal4 and LSP2-Gal4 (fat body; purple). Note that C564-Gal4 also expresses in hemocytes and some male reproductive tissues and mef2-Gal4 has been reported to express LNv neurons. (**B**) Luminescence oscillations measured from LABL flies activated by the indicated Gal4 drivers

*Figure 8 continued on next page*

*Figure 8 continued*

in peripheral tissues, in a wild-type or PDF signaling deficient genetic background, plotted over time. Colored curves represent signal from flies with no PDF signaling (*han^5304^*) compared to their wild-type counterparts (black curves). Luminescence signals from wild-type flies are reproduced in *Figure 8—figure supplement 1* for clarity. Drivers used are as indicated and described in panel A. Graphs are as described in *Figure 2C*. Location of A50 with a value close to x=0 that are not readily visible are indicated with a black triangle. (**C**) Stability of luminescence oscillations from LABL flies expressing different drivers are quantified in a wild-type (+) and PDF signaling deficient (*han^5304^*; h) background. Colors correlate with Gal4 drivers described in panel A. Color fade in the bars represent the location of the A50 calculated from the peaks and troughs. Length of bar indicates time at which luminescence oscillation becomes arrhythmic. (**D**) Differences in average period of luminescence oscillation measured by Morlet wavelet fitting, comparing wild-type (+) and PDF signaling deficient (*han^5304^*; h) flies. Colors correlate with Gal4 drivers described in panel A. Black bars represent mean period of 'rhythmic' oscillations early in measurement, +/-SEM. Red bars represent mean period of all wavelet fits (including those >48 hr). The dotted and solid horizontal lines denote 24 and 48 hras a point of reference, respectively. Periods of oscillations between *han^5304^* and wild-type flies before decay into arrhythmic or infradian oscillations (i.e. <48 hr) are averaged (black bars), and statistical significance determined between four experimental replicates (t-test). *: p<0.05, NS: not significant. (**E**) Changes in oscillation period over time determined by Morlet wavelet fitting. Colored dots represent best fits from flies with no PDF signaling (*han^5304^*) compared to their wild-type counterparts (black curves). Colors correlate with Gal4 drivers described in panel A. The dotted horizontal line denotes 24 hr as a point of reference.

The online version of this article includes the following figure supplement(s) for figure 8:

**Figure supplement 1.** Peripheral clocks exhibit ~24 hr oscillation patterns in wild-type LABL flies.

(and the 5th sLNv) regulate evening anticipation in LD and rhythmic locomotion in constant light conditions in a specific genetic background (*Murad et al., 2007*; *Picot et al., 2007*). These different responsibilities for distinct aspects of rhythmic locomotion by distinct neurons suggest that coherent behavioural rhythms are the result of converged function of the entire circadian neuronal network. It follows that distinct neuronal functions may necessitate the need for different molecular mechanisms to regulate individual clocks. Indeed, Casein Kinase II, a regulator of PER/TIM repressor complex nuclear accumulation, is detectably expressed in the LNvs but not in other circadian neurons (*Lin et al., 2002*; *Top et al., 2016*). LABL offers an opportunity to explore the effect of mutations and different environmental factors on these distinct clocks.

LABL offers several advantages in its use as a reporter in *Drosophila*. Unlike other systems that rely on measuring luminescence signal from explanted fly brains, LABL leaves the brain and other tissues in place, allowing them to remain responsive to physiological input from the body of the animal. Current techniques that remove tissues from the context of the whole-body measure clock oscillations in the absence of physiological regulatory factors. For example, explanted brains reveal ~29 hr tim-luc oscillations in contrast to ~24 hr 8.0-luc oscillations measured in a subset of clock neurons (*Versteven et al., 2020*; *Yoshii et al., 2009*). Moreover, differences in l-LNv neuronal firing are explained by the presence of eyes (*Muraro and Ceriani, 2015*), suggesting that a number of external factors maintain physiological oscillations. LABL also allows animals to roam freely, avoiding any aberrations that may arise from tethering the head of the animal to an imaging stage. However, this technique does rely on exclusive expression of Gal4 in the intended target tissues. For example, in characterizing some common circadian Gal4 lines, such as Clk4.1-, Mai179- and Clk9M-Gal4, we found that they expressed Gal4 in peripheral tissues, which disqualifies use of these drivers for neuron-specific LABL-mediated monitoring of transcriptional oscillations (*Figure 6*).

LABL is also distinct from other in vivo luminescence-based reporters. Reporters such as *plo* and PER-BG::Luc report the cumulative oscillation of either the *period* promoter or Period protein in the whole animal (*Figure 3A*). Other reporters rely on the oscillation of calcium levels (*Guo et al., 2017*) or the oscillation of secondary signaling such as NFκB or CREB2 (*Tanenhaus et al., 2012*), which measure cell responses to extracellular signals, but not the circadian clock itself. While these methods are effective in measuring how targeted tissues responds to signals from elsewhere, they do not directly measure the transcription/translation feedback loop. In contrast, LABL permits the direct measurement of the circadian clock in a target cell or target tissue, which is critical to understanding the effect of a mutation or environmental input on distinct clocks.

Using LABL, we confirmed that distinct circadian clocks exhibit distinct characteristics. We noted, for example, a sharp dip in luminescence that bisects the first peak of oscillation in all measured wild-type flies (*Figure 2C*, *Figure 2—figure supplement 1*), except with PER-BG::Luc flies (*Figure 3A*), suggesting that this may be a transcription regulation-specific phenomenon, as previously suggested by others (*Brandes et al., 1996*). We also observed that not all oscillators are maintained under constant conditions (e.g. NP3084-Gal4 activated LABL) (*Figure 8*), suggesting that a subset of clocks

(i.e. the transcription feedback loop) may at times be sustained by environmental or other forms of input. Such differences in sustaining clock oscillations have also been observed by others (*Roberts et al., 2015*; *Veleri et al., 2003*; *Veleri and Wülbeck, 2004*). For example, the oscillations of the clock protein PER are rapidly dampened in l-LNvs and DN1s in constant dark conditions (compared to other neuronal clusters), while their *per* transcript oscillations are reportedly relatively unaffected, highlighting differences between PER stability and transcription activity (*Roberts et al., 2015*; *Veleri and Wülbeck, 2004*). We too observed oscillations sustained for days in R18H11-Gal4 activated LABL flies (i.e. in the DN1s; *Figures 6 and 7*). Thus, LABL allowed oscillations to be monitored rapidly, without the need for labor-intensive, low time-resolution immunofluorescence staining or ex vivo luminescence imaging.

We were also able to employ LABL to rapidly assess the effects that loss of PDF signaling have on tissue-specific clocks as a proof of principle for this method to be used in different mutant contexts. We found that transcriptional oscillations became longer to varying degrees when circadian Gal4 lines are used to activate LABL in flies lacking PDF signaling, or shorter in the case of the fat body clocks. R18H11-Gal4 activated LABL flies exhibited no period change of transcription oscillation. Oscillations in peripheral tissue became less stable, while remaining intact. In two Gal4 lines, mef2- and esg-Gal4, circadian oscillations were lost within a few days before they were restored later in the time course, perhaps re-established through signals received from other clocks.

Loss of PDF signaling also reveals infradian oscillations of luminescence in some tissues. These oscillations were clock-dependent, since they were distinct from measurement from flies lacking a functional clock (*Figure 3B and C*). We are not certain how these infradian oscillations form, but they are unlikely to be caused by non-synchronised individual flies, since flies lacking PDF-signaling (*han5304*) revealed both ~24 hr oscillations (Pdf-Gal4 activated LABL) and ~60 hr oscillations (R18H11-Gal4 activated LABL), despite being otherwise genotypically identical. Any desynchrony in the fly population should affect these measurements in the same fashion, regardless of where LABL is being activated. These rhythms are also unlikely to be the result of feeding or differences in luciferin absorption into different tissues. If there were to be a 60 hr luminescence oscillation that stemmed from feeding cycles of luciferin supplemented food, the Pdf-Gal4 activated LABL flies would exhibit ~24 hr oscillations embedded into 60 hr oscillations. However, this is not the case. If instead we consider a metabolic explanation wherein some tissues absorb luciferin with a ~60 hr rhythm while others continue to absorb luciferin with a ~24 hr rhythm, we would need to assume the existence of a 60 hr luciferin absorption oscillation that is both dependent on the presence of a clock and absence of PDF-signaling. A third possibility is that subtle genetic differences, either among Gal4 drivers or fly backgrounds, are causing the formation of 60-hr infradian oscillations. We suggest that this is also unlikely, because flies of identical genetic backgrounds that have been aged (compared to young flies) can develop 60-hr infradian oscillations (*Figure 4*). The observed infradian oscillations are therefore unlikely to be rooted in feeding, metabolism, or genetic differences, and there must be an underlying clock-related explanation to these oscillations.

There are three possible explanations for the observed 60-hr infradian rhythms. The simplest of these explanations is based on the well-known physical phenomenon called 'beats', in which short, ultradian oscillators that are tightly coupled in phase can form circadian oscillations (*Chance et al., 1967*). For beats to produce stable infradian rhythms, the individual oscillators themselves must be held at fixed phase and period in relation to each other. Other mechanisms of producing longer period oscillations involve directly- (*Pavlidis, 1969*) or indirectly- (*Paetkau et al., 2006*) coupled faster (i.e. short-period) oscillators. It may therefore be possible to tightly couple circadian oscillators to form infradian oscillators. However, in either case this explanation necessitates that in the absence of PDF-signaling (e.g. *han5304* flies) circadian oscillators remain coupled to each other to produce infradian rhythms. With increased temperature, the phase of luciferase oscillations in 8.0-*luc* flies changes, while the period of oscillation remains the same (*Versteven et al., 2020*). If this temperature-dependent change in overall phase is caused by a change in phase coupling between distinct oscillators, as we have shown is possible (*Figure 5A*) and considering that *pdf01* flies do not exhibit this change in phase (*Versteven et al., 2020*), it is possible that a loss of PDF-signaling leads to tighter coupling between oscillators (i.e. less phase drift) to create infradian oscillations. This may also explain why amplitude in flies lacking PDF-signaling is much higher than in wild-type flies (*Figures 7 and 8*; *Veleri et al., 2003*; *Yoshii et al., 2009*). Finally, we cannot rule out that weak 60-hr oscillators exist in flies to begin

with, and become exposed as the PDF-dependent oscillators fall into arrhythmia in flies lacking PDF signaling. However, we could not uncover any evidence of such infradian oscillations using modified analysis of behavioural rhythms or by measuring PER and TIM protein levels in fly head protein extracts (*Figure 3—figure supplement 1A and B*).

The LABL reporter system promises to uncover novel clock mechanisms at increased cellular and temporal resolution. Here, we present data that demonstrate the function and use of this system. LABL has the advantage that it reveals clock oscillations directly, at the intersection of tissues (and cells) in which the *per* promoter is active (i.e. in clock cells) and Gal4 is expressed, such that cells that express Gal4 but no clock would not interfere with luminescence measurements. LABL reveals distinct clocks that are unique in their characteristics and in their responses to molecular input (e.g. PDF signaling). Together, LABL is an important new tool in directly measuring and quantifying subtle changes to individual circadian neuronal sub-types as well as peripheral clocks in *Drosophila*, in vivo, in real time.

## Materials and methods

### Fly strains

The following lines were used in the study: tim-UAS-Gal4 (A3) (*Blau and Young, 1999*), Pdf-Gal4 (*Park et al., 2000*), DvPdf-Gal4 (*Guo et al., 2014*), R18H11-Gal4 (*Guo et al., 2016*), Clk4.1-Gal4 (*Zhang et al., 2010a*; *Zhang et al., 2010b*), per$^S$ and per$^L$ (*Konopka and Benzer, 1971*), elav-Gal4 (gift from Michael W. Young); Pdfr$^{5304}$ (han$^{5304}$) (*Hyun et al., 2005*) (gift from Paul Taghert); Mai179-Gal4 (*Picot et al., 2007*) (gift from C. Helfrich-Foerster); Clk9M-Gal4 (*Kaneko et al., 2012*) (gift from O. T. Shafer); 3xUAS-FLP2::pest (*Nern et al., 2011*) (Janelia Research Campus); PDF receptor mutant han$^{5304}$ (*Hyun et al., 2005*; *Mertens et al., 2005*) (gift from P. H. Taghert); esg-Gal4 (gift from Ben Ohlstein); NP3084-Gal4 (gift from David Walker). G-TRACE flies (32251) *Evans et al., 2009*; mef2-Gal4 (27390); C564-Gal4 (6982); LSP2-Gal4 (6357) were acquired from BDSC. Genotypes of all flies used in experiments are summarized in *Supplementary file 1*. Flies were reared on standard cornmeal/agar/yeast/molasses medium at room temperature (22 °C) or 18 °C in ambient laboratory light. Flies were entrained in a light-dark cycle (12 hr:12 h) at 25 °C.

### LABL flies

LABL was constructed into an attB cloning vector for *Drosophila* embryo injection and PhiC31-mediated genome integration (*Bischof et al., 2007*; *Figure 1—figure supplement 1*). The minimal *per* promoter (*Bargiello et al., 1984*) (ranging from restriction sites SphI to XbaI) was first cloned into the pattB vector. The 5' untranslated region, ranging from XbaI site to the *per* ATG start codon was rebuilt using standard PCR: the ATG was conserved and an FRT sequence followed by a NotI restriction site was added. mCherry was amplified using standard PCR, its start codon eliminated, and a 5' NotI restriction site, three 3' stop codons and an EcoRI restriction site added. Luc2 (Promega) was amplified using standard PCR its start codon eliminated, a 5' EcoRI restriction site followed by FRT sequence added and then 3' XhoI and KpnI restriction sites added. The plasmid was then assembled and injected into embryos (BestGene Inc).

### Luminescence assay

Luminescence of flies were measured using the LumiCycle 32 Color (Actimetrics). Custom 35 mm plates (designed by Actimetrics) were used to adapt the LumiCycle 32 for *Drosophila* use. D-luciferin potassium salt (Cayman Chemicals or Gold Biotechnology) was mixed with standard fly food to a final concentration of 15 mM. Volume of food in the plate was sufficient to limit fly movement along the z-axis of the plates. 15 flies were placed in each plate and covered with a cover slip. Each of the four replicates were recorded in a different position in the luminometer to ensure equal representation of data from each of the four photomultiplier tubes. Luminescence from each plate was recorded in 4-min intervals on a standard Windows-operated PC using software by Actimetrics.

Luminescence of transfected S2 cells were measured using a liquid scintillation counter LS6000IC (Beckman) in single-photon collection mode. Cells were transfected in six-well plates at 80% confluency using Effectene transfection reagent (Qiagen) following the manufacturer's protocol. Cells were transfected with 200 ng total DNA with the indicated plasmids distributed in equivalent amounts.

Plasmids used were pAc-*Clk*-CFP (actin promoter driven *clock* fused to CFP) (*Top et al., 2018*), pAc-FLP (actin driven Flipase; gift from Nicholas Stavropoulos), LABL plasmid (above). Cells were fed 24 hr after transfection and lysed another 24 hr later in Cell Culture Lysis Reagent (Promega). Extracts were mixed with Luciferase Assay Reagent (Promega) in a 1:5 ratio and monitored for luminescence.

## Luminescence analysis

Actimetrics analysis software was used to normalize the exponential decay of luminescence signal over days using a polynomial curve fit, with no smoothing. Data were exported into.csv files for analysis. Custom python code was used to organize luminescence data into 30 min bins and quantify peaks, troughs and decay of oscillating luminescence signal (LABLv9.py; https://www.top-lab.org/downloads or https://github.com/deniztop/LABL) (*Top, 2022*). A sinusoidal curve was fitted to each day,+/-half a day, with the x- and y- coordinates for the local maxima and minima identified and quantified. Overlapping values for each peak and trough were averaged for x- and y- coordinates before being plotted,+/-standard deviation. Graphpad Prism 9 software was used to fit an S-curve to these points, and the calculated value for IC50 was used to identify the point of inflection to identify the reported A50. Custom python code was also used to quantify periods of oscillations using a Morlet wavelet fit (waveletsv4.py; https://www.top-lab.org/downloads or https://github.com/deniztop/LABL) (*Top, 2022*), as previously described by others (*Leise and Harrington, 2011*). The bottom 25% of best fits were omitted from analysis. Computer-generated oscillation data were used to confirm the parameters for analysis (data not shown): bandwidth = 3; central frequency = 1; data points in an hour = 15 (i.e. 15x4 min in an hour); zero-hour mark = 10 (10:00 am, subjective dawn); period of shortest wave match = 16; period of longest wave match = 72; increment of wavelet fit = 1; percent threshold for discarding data as a fraction = 0.25. Highest confidence interval values were used to plot period across time. The time (x-value) when an oscillation was considered arrhythmic was defined as the last 25 data points before a wavelet fit exhibited a jump from ~24 hr to ~60 hr, and averaged.

## Locomotor activity

Individual flies were analyzed for locomotor activity for three days in a light-dark cycle, followed by seven days in constant darkness using the *Drosophila* Activity Monitor System 5 (Trikinetics). Circadian behaviour was determined in the analysis of time in constant darkness. Periodicity of rhythmic locomotor activity was determined using ImageJ plug-in ActogramJ (Lomb-Scargle) (*Schmid et al., 2011*).

## Fluorescence microscopy and immunohistochemistry

To measure loss of mCherry signal from LABL plasmid in S2 cells, cells were grown in Lab-Tek II chamber slides (Nunc, Rochester, NY). Cells were imaged using a DeltaVision system (Applied Precision, Issaquah, WA) equipped with an inverted Olympus IX70 microscope (60 X oil objective, 1.42 N.A.), a CFP/YFP/mCherry filter set and dichroic mirror (Chroma, Foothill Ranch, CA), a CCD camera (Photometrics, Tucson, AZ), and an XYZ piezoelectric stage for locating and revisiting multiple cells.

Fly brains were collected at ZT23, fixed, mounted, and imaged using Leica confocal microscopy as previously described (*Top et al., 2018*; *Top et al., 2016*). Briefly, fly heads were fixed in PBS with 4% paraformaldehyde and 0.5% Triton X-100. Brains were dissected and washed in PBS with 0.5% Triton X-100. Brains were then probed with a 1:1000 dilution of rat anti-TIM (*Myers et al., 1996*), a 1:1000 dilution of chicken anti-GFP (Aves Labs), and a 1:1000 dilution of rabbit anti-mCherry (Rockland). Washed brains were re-probed using a 1:500 dilution of goat anti-chicken conjugated to Alexa-488, a 1:200 dilution of donkey anti-rat conjugated to Alexa-594, and a 1:200 dilution of donkey anti-rabbit conjugated to Alexa-647 secondary antibodies (Jackson Immunological). Brains were mounted using Fluoromount G (Beckman Coulter, Brea, CA) and imaged using a ZEISS LSM 710 confocal microscopy at ×40 magnification.

## Immunoblotting

Fly heads were collected from iso31 and *han*[5304] flies at 6-hr intervals, specifically CT03, 09, 15, and 21. Protein was extracted from fly heads as described earlier (*Top et al., 2016*). Protein samples were quantified using a Lowry Assay (Bio-Rad Cat. 5000112) and transferred to PVDF membrane using standard methods. Blots were probed with antibodies against PER (1:500) (gift from Michael W.

Young), TIM (1:3000) (*Myers et al., 1996*), and tubulin (1:20,000) (Sigma, Cat. T6199). HRP-conjugated secondary antibodies against rat (Jackson, Cat. 712-035-153) and mouse (Jackson, Cat. 715-036-151) were used at 1:10,000 dilution.

## Oscillation simulations

### Abrupt phase change

Using Matlab we simulated N=100 sinusoidal oscillators at increments of 0.5 hr each with a 24-hr period for 10 days and with no phase difference at t=0 hr. After 1.5 days, we forced half of the oscillators to abruptly phase shift by 0, 0.5, 1, 2, 4, 6, 10, or 12 hr while maintaining the same 24-hr period. *Gradual phase change*: Using Matlab we simulated N=100 sinusoidal oscillators at increments of 0.5 hr each, each with a 24-hr period for 14 days. At each time point, individual oscillator phases were randomly picked from a normal distribution with mean 0 and standard deviation, σ, that increased linearly with time from σ=0 hr at t=0 hr to either σ=6 hr or 12 hr at t=336 hr (14 days). With these parameters, oscillators had no phase difference at t=0 hr but had a maximum difference of ~40 hr (σ=6 hr) or ~100 hr (σ=12 hr) toward the end of the timeseries.

## Luminescence imaging

Flies were kept at 25 °C under light-dark cycles in vials containing 1% agar and 5% sucrose food supplemented with 15 mM luciferin for at least 3 days prior to the experiment. Images were taken using a Luminoview LV200 (Olympus) bioluminescence imaging system, equipped with an EM-CCD camera (0.688 MHz EM-CCD CAM-ImageEM X2, Hamamatsu, Japan) at a time point corresponding to the peak phase of expression (between ZT 18 and ZT 20).

Adult flies were anesthetized with ether and dry mounted on a CellView glass bottom imaging dish (35x10 mm; Greiner Bio One). Once the position of the fly was determined by bright field microscopy, bioluminescence was measured for 5 min with an EM gain of 400, using a UPLSAPO 20 x Apochromat, followed by 600ms exposure to fluorescence light. After this, another bright field image was captured and compared to the initial image, in order to ensure that the fly had not moved.

Preparation of the brains was similar to that described previously (*Schubert et al., 2020*; *Versteven et al., 2020*). Briefly, brains were dissected under red light in ice cold $Ca^{2+}$ free Ringer's solution and placed on a CellView glass bottom imaging dish (35x10 mm; Greiner Bio One) treated with Heptane glue, where the culture medium was added. The medium contains 20% heat-inactivated fetal bovine serum, 1% Penicillin-Streptomycin and 0.75 μM Luciferin in Schneider's medium (Sigma-Aldrich). Bioluminescence images were acquired with a UPLSAPO 40 x by exposing the brains for 5 min with an EM gain of 400. For each brain, a Z-stack of 13 slices was taken, each slice containing a bioluminescence, a fluorescence (200ms), and a bright field image. All images were processed in FIJI (*Schindelin et al., 2012*) and combined with bright field images in GIMP (https://www.gimp.org/).

## Acknowledgements

This work was supported by a grant from NSERC (RGPIN-2019–06101) to DT, from NSF (IOS 1656603) to SS. and from the Deutsche Forschungsgemeinschaft (DFG) (INST 211/835–1 FUGG, and STA421/7-1) to RS.

## Additional information

### Competing interests

Inan Top: is an employee of You.i Labs Inc. The other authors declare that no competing interests exist.

### Funding

| Funder | Grant reference number | Author |
| --- | --- | --- |
| Natural Sciences and Engineering Research Council of Canada | RGPIN-2019-06101 | Deniz Top |

| Funder | Grant reference number | Author |
|---|---|---|
| National Science Foundation | IOS 1656603 | Sheyum Syed |
| Deutsche Forschungsgemeinschaft | INST 211/835-1 FUGG | Ralf Stanewsky |
| Deutsche Forschungsgemeinschaft | STA421/7-1 | Ralf Stanewsky |
| Canadian Institutes of Health Research | FRN-178220 | Deniz Top |

The funders had no role in study design, data collection and interpretation, or the decision to submit the work for publication.

## Author contributions

Peter S Johnstone, Data curation, Formal analysis, Investigation, Visualization, Methodology, Writing - original draft, Writing – review and editing; Maite Ogueta, Data curation, Formal analysis, Investigation, Methodology, Writing - original draft, Writing – review and editing; Olga Akay, Data curation, Formal analysis, Investigation, Writing – review and editing; Inan Top, Software, Methodology, Writing – review and editing; Sheyum Syed, Formal analysis, Validation; Ralf Stanewsky, Resources, Data curation, Funding acquisition, Visualization, Methodology, Writing - original draft, Writing – review and editing; Deniz Top, Conceptualization, Resources, Data curation, Formal analysis, Supervision, Funding acquisition, Investigation, Visualization, Methodology, Writing - original draft, Project administration, Writing – review and editing

## Author ORCIDs

Peter S Johnstone ⓘ http://orcid.org/0000-0002-9421-811X
Ralf Stanewsky ⓘ http://orcid.org/0000-0001-8238-6864
Deniz Top ⓘ http://orcid.org/0000-0002-1042-8460

## Decision letter and Author response

Decision letter https://doi.org/10.7554/eLife.77029.sa1
Author response https://doi.org/10.7554/eLife.77029.sa2

# Additional files

## Supplementary files

• Transparent reporting form

• Supplementary file 1. Genotypes of flies used in the figures. For LABL experiments, stocks are maintained with one parental line expressing both the Gal4 driver and LABL reporter, and the other parental line expressing UAS-FLP.

## Data availability

The codes used in data analysis can be found at https://github.com/deniztop/LABL, (copy archive at swh:1:rev:67f0256ccd5a1469bf33da1a860f1c845c89b330). All data points used in generating the figures can be found at Dryad.

The following dataset was generated:

| Author(s) | Year | Dataset title | Dataset URL | Database and Identifier |
|---|---|---|---|---|
| Top D | 2022 | Data from: Real time, in vivo measurement of neuronal and peripheral clocks in *Drosophila melanogaster* | http://dx.doi.org/10.5061/dryad.2v6wwpzpj | Dryad Digital Repository, 10.5061/dryad.2v6wwpzpj |

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
