## [Editor Report]

This manuscript will be of broad interest primarily to readers in the field of Chronobiology, but more in general, also Physiology. The reporter construct generated in this study provides a great tool to dissect with cell and tissue specificity the rhythmic transcriptional oscillations orchestrated by circadian clocks in vivo. Here, the authors take full advantage of such a tool showing how neuronal and peripheral clocks might be differentially regulated and possess distinct properties.

---

## [Decision Letter]

**Decision letter after peer review:**

Thank you for submitting your article "Real time, in vivo measurement of neuronal and peripheral clocks in *Drosophila melanogaster*" for consideration by *eLife*. Your article has been reviewed by 3 peer reviewers, and the evaluation has been overseen by a Reviewing Editor and K VijayRaghavan as the Senior Editor. The reviewers have opted to remain anonymous.

There is a unanimous high interest in the technique, but a much more critical perception of the conceptual advances. Thus, we ask the authors to resubmit their manuscript under the "Tools and resource" section of *eLife*. Addressing the concerns raised towards the validity of your new technique will thus be essential, while we could allow a bit more flexibility on how you will exactly address the concerns raised towards their claimed conceptual advances.

Nevertheless, it is clear that what you will claim as conceptual advances will also need to pass the scientific quality checks.

With this in mind, please address each of the four essential revisions points below. Please also carefully read the individual reviews, as they have been written with great attention to detail and in a highly constructive attitude. While it won't be necessary to address them point-by-point, we believe that they can be helpful to improve your manuscript.

Essential revisions:

1) Better analysis, visualization and further validation of the overall technical approach.

The further validation should include additional experiments, such as the test of circadian clock mutants that slow down/speed up the clock or direct measurements of clock protein/RNA levels.

2) Further validation of the usability of the inflection point of rhythmicity. The reliability of this marker is particularly critical as the interpretation of the PDFR data relies largely on it.

We suggest performing an experiment to demonstrate how the inflection point changes according to clock stability with age. It has been shown in both flies and mice that older individuals exhibit a less stable oscillator. The authors could test their assumption monitoring oscillations in young vs old flies.

3) A critical and careful discussion of the PDFR results in the context of previous work, that clearly explains which results are consistent with or challenge previous models. (See several points of critique about the lack of proper referral to the existing scientific literature in the individual reviews.)

4) There have been several questions/concerns raised concerning the reported 60hr rhythms, such as

– Weak rhythms such as the infradian rhythms need to be better visualized.

– Is there additional evidence (experimental or theoretical) that these are really 60hr rhythms or could these rather represent relatively randomly occurring superpositions of peaks/throughs of de-synchronized cells in tissues? Potentially consistent with the latter hypothesis it appears that these rhythms exhibit a rather large period range (between 48-72hrs)?

– Could such infradian rhythms be artificially driven by feeding or other metabolic rhythms that affect the amount of luciferin in the body?

*Reviewer #1 (Recommendations for the authors):*

– Currently, a genetically encoded reporter cannot be exploited to assess specific aspects of human diseases for ethical reasons, although in the future this might be possible. Thus, I would strongly encourage the authors to be (1) more focused on the short-term application, and/or (2) more specific (cell culture, organoids, monitoring disease progression but in animal models, etc…). The authors used many references (especially in the introduction) to highlight medical application, but this, in my personal opinion, has to be moderated as such approach would be limited by the current regulations in humans (gene therapy is a different aspect, it is not used to monitor specific aspects of our biology but to potentially improve pathological conditions).

– The authors based their analyses on the identification of the inflection point, considered a readout for clock stability. I would suggest performing an experiment to demonstrate how the inflection point changes according to clock stability. For instance, in both flies and mice, it has been shown that older individuals exhibit a less stable oscillator. The authors could test their assumption monitoring oscillations in young vs old flies. The experiment performed in tim01 background is valuable, but of course in this case the oscillations are severely disrupted. If the inflection point is a true readout, we should expect this to occur later and earlier in young and old flies, respectively. Concerning this point, I am also open to other suggestions from the authors, but I really think that this assumption, although logical from a certain point of view, should be experimentally validated. On the same line, it would make the data more robust by having a demonstration that the inflection point is not influenced by the genetic background (i.e. the GAL4 strains used might differ from these points of view).

– One of my biggest concerns revolves around the experiments and the conclusions on PDF signaling. First, I did not fully understand why the authors chose to perform their experiments in pdfr-/- (han-/-) background instead of pdf-/-, especially knowing that PDFR might be responsive also to other neuropeptides (i.e. DH31) (an aspect correctly reported by the authors). Yet, most importantly, I am not convinced their conclusions about the negligible effects of PDF signaling on peripheral clocks are supported by the experimental plan used. To my knowledge, pdfr has never been shown as expressed in gut, fat bodies, and muscle (flybase reports expression in head and crop). More in general, PDF signaling has not been widely considered to synchronize/coordinate peripheral clocks (outside the head and prothoracic gland). This is a plausible assumption, but without data confirming the expression of pdfr in the tissues included in the analyses, no conclusion based on PDF signaling effects in these peripheral tissues can be drawn (i.e. "This result suggests that the presumed dominance of the brain clock in regulating peripheral clocks needs to be re-examined"). Thus, I encourage the authors to present evidence for such expression pattern (i.e. using immunohistochemistry), if this has not already been reported in the literature.

– My second major concern is about the lack of investigation and explanation of the ~60 h oscillatory dynamics identified. I understand that describing the ~60 h infradian oscillations goes beyond the scope of the study for the authors, yet, this represents such an atypical rhythm that I think the authors should provide a biological explanation about it, at least at the hypothesis/theoretical level. Moreover, I suggest discussing the findings concerning infradian rhythms unveiled in constant conditions in the context of the existing literature on ultradian and infradian rhythms unmasked by DD, LL, etc… On the other hand, graphs suggest infradian oscillations are actually very broad compared to the circadian ones (which are precisely around 24 hours) and range from ~48 h to ~72 h. Thus, I think it is not appropriate to talk about ~60 h rhythms. Moreover, this broad rhythmicity identified at the luminescence level might be of little to no biological meaning, being simply asynchronous, noisy, and stochastic oscillations. Maybe the authors could test whether this rhythmicity is present at the locomotor activity level as well. I think this would demonstrate that this infradian rhythmicity has a real link with *Drosophila* biology. However, also if this was the case, could we really consider it meaningful rhythmicity in constant conditions?

*Reviewer #2 (Recommendations for the authors):*

This manuscript describes a novel intersectional approach (LABL) that allows to express a per-luciferase driver in specific cell types or tissues in *Drosophila*. This is definitely a welcome addition to the arsenal of tools to study *Drosophila* circadian rhythms. Overall , the authors make a strong case that their approach works well. They are even able to record from a few brain neurons. The authors then use their novel method to test the impact of loss of PDF-receptor (PDFR) signaling on brain and peripheral oscillators. There, I have difficulties with the analysis and interpretation of the data, particularly when it comes to defining some form of hierarchy between circadian pacemaker neurons and peripheral tissues. There is already considerable literature on this, but the authors need to better link their results to current knowledge, and draw clear conclusions on whether their results indeed reveal something new. Results are not always coherent, and I am concerned that one of the main measurements used in these studies, called "amplitude stability constant", is not a reliable method to study the decay in rhythmicity. In summary, this paper brings a novel technology to the study of circadian rhythms in *Drosophila*, but its use to study PDFR signaling across fruit flies seems to provide rather limited new insight into the hierarchy between brain and body clocks, at least as currently presented.

Specific comments:

1) In the abstract, the authors say their results challenge the notion that brain clocks are dominant over peripheral clocks. However, it has been quite clear for a long time that peripheral clocks in *Drosophila* function independently. Most have their own photoreceptors, and do not depend on the brain for their oscillations under LD and for a few days at least in DD. There are a few exceptions though. An interesting one is the oenocytes, which receive PDF signaling (Krupp 2013). However, even there, rhythm amplitude is not affected by loss of PDF signaling, only phase. The Prothoracic Gland is however dependent on sLNv signaling for its rhythms, but this is through sNPF (Selcho 2017). The authors need to carefully survey relevant literature and place their results in context. It would be really interesting to try an oenocyte-GAL4 line and test whether LABL can pick up changes in phase of rhythms in the absence of PDFR in oenocytes. This would be a nice validation of this new approach.

2) In the introduction, the authors mention the lack of tissue-specific methods to monitor circadian rhythms in real-time. In flies, such methods are indeed lacking, but not in mammals. Intersectional approaches have been developed by the Schibler (Sinturel 2021) and Weaver (Smith 2022) labs, and those should be mentioned in the introduction. Sinturel should also definitely be mentioned in the discussion as their results are very directly relevant to the present study.

3) The S-curve fitting of peak and through to determine rhythm decay does not appear reliable to me. This approach is going to be very sensitive to experimental noise, given how flat these curves are. Several curves do not look like an S (see Figures 5 and 6). How reproducible were these decay measurements across experiments? There are no statistics on Figure 5B and 6C. I am also not sure that the term "amplitude stability constant" is appropriate. This is not a constant.

4) I am not convinced that the infra-dian oscillations are absent in tim0 flies, just that they are less robust or synchronized than in pdfr-han. These oscillations might be the result of a weak underlying behavioral rhythm such as feeding, which could lead to low amplitude cycles of luciferin in the body. The weak inverted rhythms that appear after a few days in the esg-GAL4 experiment on Figure 6 could also be due to rhythmic ingestion and could explain the inversion of oscillations.

5) In the image for luciferase signal in the LNvs on Figure 4C, the signal seems to be too dorsally located to be from the LNvs. Even in the DvPDF image, the localization seems too dorsal. Would there be a luciferase antibody that could be used, to directly visualize LUC expression by ICC? The G-trace images look good, but they do not show directly luciferase expression.

6) On page 11, the authors seem to conclude that body clocks need PDFR signaling, which contradicts one of the abstract's main statements. Then, the results in figure 6 seem rather different. Overall, loss of PDFR signaling does not seem to have much effect on figure 6 in peripheral clocks. How are those results reconciled?

[Editors’ note: further revisions were suggested prior to acceptance, as described below.]

Thank you for resubmitting your work entitled "Real time, in vivo measurement of neuronal and peripheral clocks in *Drosophila melanogaster*" for further consideration by *eLife*. Your revised article has been evaluated by K VijayRaghavan (Senior Editor) and a Reviewing Editor.

The manuscript has improved significantly and all reviewers agree that it is now in principle suitable for publication. However, before we can finally accept your manuscript we would ask you to address some remaining issues, as outlined below:

There are still some concerns on the interpretation of the infradian rhythms from reviewers 1 and 2 that we would ask you to respond to.

Please provide better images of the luciferase signal in the brain of the pdfGAL4 and DvpdfGAL4 flies.

Finally, please respond to the specific comments raised by reviewer 2, as these point out several mistakes or sentences that might need rephrasing for clarity.

For your information, please find the full reviews below.

*Reviewer #1 (Recommendations for the authors):*

I personally consider the manuscript entitled "Real time, in vivo measurement of neuronal and peripheral clocks in *Drosophila melanogaster*" significantly improved overall. The authors conducted many suggested experiments and strengthened at the same time the weak points in the interpretation of generated data, phrasing, and literature gaps.

The authors tackled with appreciated effort all the 4 major points of concern raised by the reviewers. Particularly, the authors effectively answered (a) my criticism concerning the misleading statements on the applicability/potential of the methodology developed in disease contexts, and (b) all the points I raised in my initial review.

Concerning my criticism of the inflection point, the authors definitely strengthened the reproducibility of their analysis but failed to provide convincing evidence on clock stability in aged flies and the effect of the genetic background. However, I appreciated the dedication and effort in the experiments I suggested. I understand the failed aged flies` experiment is caused by insurmountable technical aspects, and I am satisfied with the transparency they used in openly addressing the genetic background issue I raised.

Similarly, I positively value how the authors addressed my concerns regarding their interpretation of the role of PDF signaling. I understand the "genetically-convenience" reasons (considering their major goal), and the motivations for why they did not directly tackle PDFR expression, and I appreciate the implemented changes in the phrasing of some potentially misleading sentences.

Concerning the ~60 h rhythm concerns, although the experiments did not provide positive evidence, I again appreciate the authors´ investment in experimentally addressing these criticisms. At the same time, from a reader standpoint, the manuscript transparency is significantly improved, thanks to the inclusion of these data as supplementary information, and the theoretical explanations provided in the text. To be completely honest with the authors, I still feel the genetic background might have a bigger role than we think, yet, I do think the oscillator coupling argument is very valid and definitely the most plausible hypothesis, and I share their enthusiasm about future studies in this direction.

Finally, I agree with the editors about moving the manuscript to the "Tools and Resource" section of *eLife*. As much as the authors conducted interesting/impactful research and careful analyses, I still feel the advancement provided in the manuscript from a biological standpoint is still surpassed by the potential of the methodology the authors developed.

All considered I endorse the publication of this manuscript on *eLife*.

*Reviewer #2 (Recommendations for the authors):*

The authors have improved their manuscript, though some issues remain. I will first go through the four main issues that were brought up in the first round of reviews.

1) Validation of the approach: This has certainly been improved with the use of clock mutants and old flies. LABL should prove to be a potent new tool for studying circadian rhythms in *Drosophila*.

2) I still have concerns with the reliability of the inflection point. It is reassuring that the authors observed a similar A50 in two independent sets of experiments done with wild-type flies. However, they acknowledge that there is important variability between individual experiments. I am not sure that inflection points will prove reliable beyond robustly rhythmic genotypes. For example, in figure 7, it does not appear that perL can be fitted with an S shape. Also, by not being able to use individual flies or even individual experiments, there is no way to test differences between genotypes with statistical analysis. Some of the differences that were observed might not hold in future experiments. The authors should acknowledge those important limitations.

3) Discussion of previous results on PDF and PDFR signaling has been improved, but I noticed issues with referencing that will be detailed below.

4) The lengthy response of the authors to the questions about the infradian rhythms is at times difficult to follow. In particular, I do not understand why they exclude some form of metabolic/feeding rhythm. These rhythms might simply be revealed when circadian control is lost or severely impaired. They might be more clustered around 60 hrs in han mutant flies compared to tim0 , because tim0 flies have no circadian clock at all.

*Reviewer #3 (Recommendations for the authors):*

The authors have done a good job in responding to my previous critiques. They have added new experiments, new analyses, new model simulations, and new text. The changes to figures have made them clearer. While it remains unclear to this reviewer how the 60-hour LABL rhythms arise or what they mean, the discussion of possible explanations is thoughtful and the source/meaning of these infradian rhythms remains outside the scope of this paper. The genetic tools generated in this paper have been well validated and characterized and will be appropriately published in the Tools and Methodology section of *eLife*. Without a doubt, these tools will be highly useful for *Drosophila* researchers in circadian biology and this manuscript is likely to be well cited.

---

## [Author Response]

Essential revisions:1) Better analysis, visualization and further validation of the overall technical approach.The further validation should include additional experiments, such as the test of circadian clock mutants that slow down/speed up the clock or direct measurements of clock protein/RNA levels.

We have conducted further experiments to better analyze, visualize and validate our overall technical approach in four ways, using (1) the perS mutant (loss of DBT phosphorylation site), (2) the perL mutant (low PER protein expression), (3) aged flies compared to young flies and (4) measuring protein levels. These data confirm that LABL is a useful tool that allows for the measurement of clock oscillations in target cells and tissues and is versatile in its use.

(1 and 2) We have added a text section to the results entitled “Circadian clock mutants perL and perS reveal features of clocks in distinct neurons.” and also added Supplemental Figure 5, which illustrate these data, pp 14-16 in manuscript.

Our data reveal that each of the different neuronal clusters tested indeed shorten or lengthen clock oscillations as measured by our LABL method. We also demonstrate that clock oscillations in a perL genetic background are dampened, as previously reported by Rutila et al., 1996 (cited in revised manuscript).

Further to this, we once again find that each of the neuronal clusters exhibit different rates of amplitude decay, and slight variability in their period, further validating our overall technical approach.

(3) We measured luminescence oscillations in aged flies. Aged flies are known to lose robustness in their behavioural rhythms (and Supplementary Figure 4). Using LABL in aged flies, we demonstrate that clock oscillations in distinct cells oscillate with a low amplitude, reflecting the lost robustness observed in behavioural rhythms. Further to this, luminescence oscillations in aged flies exhibit no decrease in amplitude over time (as observed with young flies) or the oscillations become infradian. The infradianlike oscillations prevent us from fitting an S-curve to the peaks and troughs to measure the A50 (point of inflection of the S-curve), but also demonstrate that infradian oscillations can appear in aged flies in genetic backgrounds that don’t exhibit infradian oscillations in younger flies. This validates the observed infradian oscillations, but also suggests that genetic background is less likely to be a factor in our observation of infradian oscillations. Thus, our method of luminescence measurement describes a second biological phenomenon (age-caused decay of clock oscillations), further validating our overall technical approach. We added an additional figure (Figure 4 and Figure S4) and text (pp. 10-11) to describe our results. We also expanded our discussion to reflect on this data.

(4) We attempted to visualize 24-hour protein oscillations or decay into 60-hour infradian rhythms using protein extracts from fly heads and probing for PER and TIM. We demonstrated that we can observe PER and TIM oscillation across 24 hours, as expected. However, immunoblotting proved too insensitive to observe the amplitude decay that reveal the infradian rhythms we observe with luminescence, in a han5304 genetic background: since changes in amplitude that we observe in our luminescence oscillations were low, and since immunoblotting requires collection of different samples across time at low time resolution (i.e., one data point is not dependent on the previous data point and is collected independently), we concluded that we did not have the time resolution necessary to observe infradian rhythms. We added Supplementary Figures 2B and 2C and text (pp. 9-10) to reflect our results.

Overall, we believe that our data shows that LABL-mediated luminescence oscillations (1) predictably shorten or lengthen in different mutant backgrounds, (2) reveal low-amplitude oscillations in aged flies, as expected, and (3) correlate well with TIM and PER protein oscillations across 24 hours as revealed by immunoblot.

2) Further validation of the usability of the inflection point of rhythmicity. The reliability of this marker is particularly critical as the interpretation of the PDFR data relies largely on it.We suggest performing an experiment to demonstrate how the inflection point changes according to clock stability with age. It has been shown in both flies and mice that older individuals exhibit a less stable oscillator. The authors could test their assumption monitoring oscillations in young vs old flies.

The question regarding the reliability of the inflection point is a good one, and we thank the reviewers for encouraging us to test this.

We conducted an additional locomotor behaviour experiment with aged flies to demonstrate that in our hands aged flies also exhibit weakened locomotor rhythms (Supplementary Figure 4).

We repeated the LABL experiment in Figure 2C when conducting the experiment for aged flies and comparing them to young flies. We show that LABL flies tested for luminescence oscillations reveal nearly identical points of inflection in the new experiment using the tim-UAS-Gal4 driver: compare Figure 2C and Figure 4A. We believe this speaks to the reproducibility of the point of inflection. We added the text “We note that in young flies, the A50 fell to 6.5-7 days, almost identical to our first observation (Figure 2C), reflecting the reproducibility of this calculation.”, p. 11 to reflect this. We additionally refer the reviewers to point 4 below, under “Experiments, 3. Oscillation Simulations”. Here we describe how the amplitude decay may occur and how it fits into an S-curvelike decay.

Testing aged flies is a very good suggestion, which we have done. Our data reveal that the immediate low amplitude of older flies correlates well with the amplitude decay observed in young flies after many days in DD, as show in Figure 4. Importantly, we do not observe a decay of amplitude in which we could fit an S-curve, either because the number of data points (fitted peaks and troughs) were too few, or the data points appeared to already be decayed. In an unexpected result, we found that Pdf- and R18H11-Gal4-driven LABL flies exhibited rapid decay into infradian rhythms, which is not observed in younger flies.

We believe that this experiment validates two things. (1) The decay of amplitude that we define is indeed reliable (and reproducible) and that this decay is not observed in older flies. We assume this is because the decay has already happened earlier in the flies’ lifetime, or is not observable in our experimental setup. (2) Infradian oscillations are observed in clocks in tissues in older flies that are not observed in younger flies, suggesting that the observed infradian rhythms are not an artefact of genetic background, driver, etc., but can be age-dependent (expanded further below, point 4).

Additionally, we conducted simulations of phase uncoupling of simulated oscillators. We demonstrate that abruptly uncoupled oscillators do not exhibit changed period, only changed amplitude (Figure 5A). However, if phase uncoupling is allowed to grow randomly and linearly over time, we find that we can reproduce the amplitude decay that we observe in our data without affecting period, once again. In our simulations, the S-curve fit for peaks and troughs eventually approaches zero, because all oscillators are being uncoupled. In our experimental data (e.g. young flies in Figure 4), the bottom part of the Scurve always approaches a non-zero value, suggesting that either some oscillators do not phase shift relative to each other, or the phase shift does not continue indefinitely as it does in our simulations.

Overall with our simulations, we have shown how the decay of amplitude may occur, and we further support the hypothesis that infradian rhythms must indeed be a biological phenomenon (expanded below, point 4).

We have included additional text (pp. 11-13) and include a new figure (Figure 5) to describe our results.

3) A critical and careful discussion of the PDFR results in the context of previous work, that clearly explains which results are consistent with or challenge previous models. (See several points of critique about the lack of proper referral to the existing scientific literature in the individual reviews.)

We thank the reviewers for this point. We have expanded our discussion of our results to include appropriate context of PDF signaling and PDFR from previous work; we highlight where our data is consistent with, or challenges earlier work and we have expanded our references in support of some of our data and conclusion.

Since our initial submission, Li et al., have published a paper that describes all detectable transcripts that are expressed in various organs in flies. They report the expression of *pdfr* in a number of peripheral organs. We have cited this paper and incorporated it into our results (p. 19). We now refer to this paper as evidence that some peripheral organs express *pdfr* mRNA, but we underscore that mRNA expression does not reflect how much PDFR protein (or whether an effective amount of PDFR) is expressed.

One thing that we also would like to draw attention to is that *pdfr* expresses different spliced variants. Although Im et al., (2010, J Comp Neuro) developed some antibodies against PDFR, it is likely that this antibody is limited in its detection of different PDFR variants. Indeed, it appears that flies express three alternatively spliced versions of PDFR, which complicates this endeavour (Paul Taghert, personal communication, SRBR 2022). Unentangling which PDFR is expressed in which tissue will require considerable effort and is being undertaken by the Taghert group.

In the context of PDFR expression in peripheral organs, we also highlight our observation of a change in phase of oscillation after a few days in constant darkness in some of the peripheral tissues we tested for luminescence oscillation (i.e., fat body and muscle). We discuss how our measurements suggest that PDF plays a role in maintaining phase coupling in these tissues, and also discuss how this may be analogous to the PDF influence on coupling between cry- and cry+ LNds (pp. 20-21) (as described in Yoshii et al., 2009).

We now have added an additional section in which we underscore the differences between luminescence oscillations revealed by LABL and other systems that use universal luciferase reporters (i.e., 8.0-luc) (pp. 18-19). This is in addition to Figure 3A. Specifically, we point out that we observe longer luminescence periods in han mutants, while others observe shorter periods using 8.0-luc in pdf01 flies (p. 18) (Versteven 2020). We discuss that this difference may be due to 8.0-luc expression in a select number of neurons in the fly (Veleri et al., 2003). Despite this, we observe shorter behavioural rhythms in the rhythmic han5304 flies, similar to others and to the shorter behavioural rhythms observed in rhythmic pdf01 flies (Renn et al., 1999). Related to this, we also cite and highlight that pdf01 and han5304 flies have been shown to exhibit differences in phase advance and phase delay in fly behaviour (Krupp et al., 2013), which is a possible explanation for some of the discrepancies between our luminescence data and other published luminescence data.

We corrected an issue where we inadvertently called han5304 (and pdf01) mutant flies arrhythmic. They are *mostly* arrhythmic, not completely. We also expanded these sections to highlight that the flies that carry these mutations exhibit shorter behavioural rhythms. (pp. 8) (and previous point)

We introduce the concept of neuronal synchrony (and coupling) in the context of PDF (Lin et al., 2004 and Yoshii et al., 2009) (p. 13). We also described how pdf may work to keep some oscillators uncoupled and highlight how cry- and cry+ LNds become phase coupled in a pdf01 genetic background fly (Yoshii et al., 2009). We also suggest that the increased amplitude of luminescence oscillation that we observe in han5304 flies (and the same observation in pdf01 flies described by others), compared to wild type flies, may be a result of clocks that are more tightly coupled when PDF-signaling is eliminated (pp. 13, 16, 18 and 20).

Additionally, we have included other references throughout the manuscript, listed below:

LNvs and LNds continue to oscillate in the absence of pdf (Lin et al., 2004, Yoshii et al., 2009) (p. 17).

s-LNvs have enhanced tim promoter activity in the absence of functional PDFR (han5304) (Mezan et al., 2016) (p.17).

Pdf01 exhibits shorter behavioural period and advanced LNd nuclear translocation (Lin et al., 2004, Renn et al., 1999) (p.18).

We have also included additional references suggested by the reviewers, (e.g., oenocytes, pp 18-19).

4) There have been several questions/concerns raised concerning the reported 60hr rhythms, such as– Weak rhythms such as the infradian rhythms need to be better visualized.– Is there additional evidence (experimental or theoretical) that these are really 60hr rhythms or could these rather represent relatively randomly occurring superpositions of peaks/throughs of de-synchronized cells in tissues? Potentially consistent with the latter hypothesis it appears that these rhythms exhibit a rather large period range (between 48-72hrs)?– Could such infradian rhythms be artificially driven by feeding or other metabolic rhythms that affect the amount of luciferin in the body?

The question of what these infradian rhythms are is a good one, and we thank the reviewers for pushing us to dig deeper into this. We have conducted three experiments, including two different simulations, included citations from theoretical physicists to explain this phenomenon and make five theoretical arguments about the probable explanation for this phenomenon.

Additional Experiments:

1. Immunoblotting:

We measured PER oscillations across 6 days at 6-hour resolution by immunoblot, to reveal 60-hour rhythms. First, we extracted protein from whole flies, since our 60-hour data relied on luminescence signal from the whole flies (TUG driving LABL) (Supplementary Figure 2C). Arrhythmic PER expression in ovaries prevented us from demonstrating PER oscillation in these protein extracts from a mixed-sex fly population (Hardin, 1994 MCB). We noticed that in the SDS-PAGE wells, we were left with lipid-like impurities during the experiment, and we reasoned that these impurities may be interfering with the immunoblotting.

We therefore next extracted protein from fly heads in a reattempt of the experiment. We were able to demonstrate an ~24-hour period in the first day of constant darkness in han5304 mutant flies, which we were expecting and had shown to be the case in both locomotion and luminescence oscillations in timUAS-Gal4 driven LABL flies (Figure 3C). However, we lost reliable and reproducible patterns of PER expression at later time points. We concluded from this that immunoblotting may not be sufficiently sensitive or quantitative enough to reveal the subtle amplitude changes that would reveal a lowamplitude 60-hour period. Even if we had been successful, there would still be reason for caution: it has been reported that clocks in heads and bodies can differ in their oscillation (Hardin et al., 1994 MCB). We therefore include our new immunoblot data in the paper to demonstrate that while we are able to detect oscillations of PER and TIM protein in wildtype flies and in the first day of han5304 mutant flies, we are not able to reveal a 60-hour rhythm in these data.

2. Locomotor Behaviour:

We conducted additional locomotor behaviour experiments to try to measure 60-hour oscillations of clocks through another method means. We thank the reviewer who suggested looking for this period in locomotor activity in wt and han5304 flies. We conducted this experiment using wild type flies (24hour luminescence rhythms), han5304 (eventual 60-hour luminescence rhythms) and tim01 as a control (no rhythms). Behavioural period of all flies was measured in a 16 – 32-hour range or 48 – 72-hour range (Supplementary Figure 2A), without consideration of whether the actograms were arrhythmic (visually or by power of fit in a Lomb-Scargle method of generating a periodogram). Wild type flies demonstrated a behavioural period just under 24 hours, as expected, with a Power of ~800. han5304 flies exhibited an average of ~24-hour behavioural period with wide variability of behavioural period, and dramatically lower Power. (We would like to point out that we used all living flies to estimate behavioural period, regardless of whether or not they were arrhythmic by eye or by low Power. This may explain the discrepancy between the ~24-hour behavioural period here and slightly short behavioural periods reported in “rhythmic” flies.) tim01 flies showed high variability of behavioural period fit with no clear clustering and a very low Power. This demonstrates that overall wt flies are rhythmic, tim01 flies are arrhythmic and some han5304 flies retain weakly rhythmic behaviour.

The same data was analyzed for behavioural period fits ranging between 48 – 72 hours (Supplementary Figure 2A). Both wild type and han5304 flies demonstrated some clustering of behavioural period into two ~60 and ~50-hour clusters, while tim0 demonstrated a uniform range of variability of fits, suggesting that in wild type and han mutants there may be a behavioural rhythm that fits within a 50 – 60-hour period. However, the Power of these fits was very low, similar to that observed for tim01 flies. We concluded that these data are not reliable enough to draw conclusions from. We include these data in the manuscript as Supplemental Figure 2A to inform readers that we explored identifying a 60-hour behavioural rhythm.

3. Oscillation Simulations:

We simulated oscillations in which oscillators were abruptly uncoupled or gradually uncoupled in an attempt to produce infradian rhythms. While this simulation did not reveal infradian rhythms, it did demonstrate that phase changes of *some* oscillators cause a decay in amplitude (and overall phase). In other words, clock uncoupling and phase change does not explain the 60-hour oscillation (Figure 4) but does explain amplitude decay. The abrupt changes in phase of some oscillators caused an immediate change in amplitude and a change in phase of the total signal, whereas a gradual shift in phase of some oscillators cause a decay of amplitude that tracked an S-curve. We therefore conclude that phase uncoupling would only lead to a decline of the signal while maintaining constant period, even if the periods of oscillation were slightly varied (e.g., 23-hour periods and 24-hour periods in the same animal). In our biological system, the amplitudes of luminescence oscillation in the top end and bottom end of the S-curve are steady, declining to a stable but non-zero value. We therefore conclude that either some clocks do not change phase amongst each other, or the amount of phase change is limited (i.e., the phase can drift, but stops at a given value, unlike in our simulations where it is allowed to drift indefinitely).

Theoretical Arguments:

Refutation or support of possible explanations for the 60-hour infradian argument.

1. The Population Argument:

It has been raised that flies within the population that we are testing may “uncouple” from each other, causing a phase drift that creates the amplitude decay or infradian rhythms. Our simulations suggest that this would lead to an amplitude decay, rather than a period change. However, population uncoupling is still very unlikely because we see 24-hour rhythms in Pdf-Gal4 driven LABL han5304 flies and 60-hour rhythms in R18H11-Gal4 driven LABL han5304 flies. Since the flies are otherwise genetically identical (except for their Gal4 driver), any desynchrony in the population should affect both measurements the same. Thus, this is a highly unlikely explanation for infradian rhythms.

2. The Metabolism Argument:

Feeding: It has been suggested that feeding patterns may influence the formation of 60-hour infradian rhythms. We feel this is unlikely. If there were to be a 60-hour feeding oscillation, the Pdf-Gal4 activated LABL han5304 flies would exhibit ~24-hour oscillations embedded into 60-hour oscillations. This is not the case, suggesting that feeding rhythms do not contribute to the 60-hour infradian oscillations.

Luciferin absorption: A second possibility for a metabolism explanation is that some tissues absorb luciferin with a ~60-hour rhythm while others continue to absorb luciferin with a ~24-hour rhythm. However, this explanation creates the requirement of a 60-hour luciferin absorption oscillation instead of a 60-hour clock oscillation that also must depend on both the absence of PDF signalling and presence of a clock. Furthermore, this argument moves the 60-hour oscillation problem away from a clock-mechanism and places it on a clock-dependent luciferin absorption mechanism, “kicking the can down the road”. Importantly, it assumes that an endogenous 60-hour oscillation must exist in the background (see below, oscillator coupling argument).

3. The Genetic Background Argument:

It is also suggested that subtle genetic differences in fly backgrounds may underlie the development of 60-hour infradian oscillations. We also suggest that this is unlikely, because flies of identical genetic backgrounds that have been aged (compared to young flies) develop 60-hour infradian oscillations as well (Figure 4C). While we cannot rule out that the driver-dependent subtle genetic backgrounds combined with age may cause the development of infradian rhythms, we suggest that this is an unlikely explanation.

4. The Oscillator Coupling Argument:

A few theoretical papers describe how short-period ultradian rhythms, tightly coupled, could create 24hour oscillations. By extension, it is possible that multiple ~24-hour oscillators could create a longerperiod oscillation if they are coupled more tightly in the absence of PDF signaling. This possibility requires that PDF in wild type flies keeps some clocks uncoupled (or enforces weak coupling), which would suggest an as-yet unknown function of PDF. Versteven et al., (iScience, 2020) show change in phases of oscillation in luminescence signal in a temperature-dependent manner. The same authors observe this phase change (and amplitude reduction) disappear in a pdf01 genetic background. If increased temperature is what is uncoupling some clocks in this study, and since our simulation data suggests that abrupt phase uncoupling of oscillators can cause overall phase changes in total measured oscillators without affecting period (Figure 5A), it is possible that PDF indeed decreases the strength of coupling between oscillators (i.e., pdf01 increases coupling strength). If loss of PDF signaling causes stronger coupling between clocks, this may both explain the 60-hour infradian oscillation as well as increased amplitude of luminescence oscillation. An additional observation by Yoshii et al., (2009) demonstrates that cry- LNds normally oscillate in reverse phase with cry+ LNds – only to become synchronized (coupled) when PDF is lost. This once again suggests that PDF may indeed function to uncouple some oscillators, causing tighter coupling when it is lost, leading to longer infradian periods of luminescence oscillation.

It is possible that shorter behavioural periods are observed in flies lacking PDF, either because of differences in coupling in locomotor-specific circuits, or because shorter locomotor periods are the result of exclusion of arrhythmic flies with low Power that biases the data in one direction (see 2. Locomotor Behaviour).

5. The Endogenous Infradian Clock Argument:

It is possible that 60-hour oscillators already exist in damped form in flies. This possibility would suggest that these infradian rhythms appear when the 24-hour clocks become uncoupled when PDFsignaling is lost, decaying rapidly, and exposing these infradian oscillations. We offer this as a possible, but unsatisfying scenario for explaining the observed 60-hour infradian oscillations.

We added text to the discussion describing these possibilities, pp. 25-26.

We also added the statement “We also point to seemingly long-period oscillations observed in the DN1s, in constant light conditions in a *cry*^b^ genetic background (Nave et al., 2021). Although this mechanism is likely different, it is another example of long-period oscillations in the DN1s, which we observe in a han5304 background in DN1s after 6 days of constant darkness (Figure 6D).”, to the Discussion, to underscore a biological precedent for long-period oscillations in some neurons under specific conditions p 26.

We once again thank all three reviewers for their constructive and helpful criticisms, and we feel that they have helped us strengthen the manuscript considerably.

We would also like to address the specific points raised by the reviewers below. Where relevant, we refer the reviewers to our explanations above.

Reviewer #1 (Recommendations for the authors):– Currently, a genetically encoded reporter cannot be exploited to assess specific aspects of human diseases for ethical reasons, although in the future this might be possible. Thus, I would strongly encourage the authors to be (1) more focused on the short-term application, and/or (2) more specific (cell culture, organoids, monitoring disease progression but in animal models, etc…). The authors used many references (especially in the introduction) to highlight medical application, but this, in my personal opinion, has to be moderated as such approach would be limited by the current regulations in humans (gene therapy is a different aspect, it is not used to monitor specific aspects of our biology but to potentially improve pathological conditions).

We agree with the reviewer. We did not mean to imply that this reporter can or should be used in humans. We have made the following changes to clarify this confusion:

p.3 “Such a broad variety of pathologies associated with compromising circadian rhythms suggests a need for cheap and effective ways to measure tissue-specific circadian clocks directly in model organisms.” The sentence was changed to include the phrase “in model organisms”.

p.4 “Given that circadian clocks appear to be linked to a wide range of physiologies, including metabolism, as well as various behavioral disorders, there is a need to monitor distinct cell- and tissue-specific circadian clocks directly, in vivo, and in real time in model organisms.” The sentence was changed to include the phrase “in model organisms”.

p.5 “We believe that this technology will be critical in the interrogation of distinct circadian clocks in future studies, particularly in monitoring peripheral clock function during disease progression in *Drosophila*.” The sentence was changed to include “in *Drosophila*”

p.22 In the sentence “Using LABL, we were able to rapidly assess the effects of loss of PDF signaling on tissue-specific clocks as a proof of principle for this method in different mutant contexts.” We removed the phrase “or disease contexts” at the end of the sentence to avoid confusion regarding application to human disease.

– The authors based their analyses on the identification of the inflection point, considered a readout for clock stability. I would suggest performing an experiment to demonstrate how the inflection point changes according to clock stability. For instance, in both flies and mice, it has been shown that older individuals exhibit a less stable oscillator. The authors could test their assumption monitoring oscillations in young vs old flies. The experiment performed in tim01 background is valuable, but of course in this case the oscillations are severely disrupted. If the inflection point is a true readout, we should expect this to occur later and earlier in young and old flies, respectively. Concerning this point, I am also open to other suggestions from the authors, but I really think that this assumption, although logical from a certain point of view, should be experimentally validated. On the same line, it would make the data more robust by having a demonstration that the inflection point is not influenced by the genetic background (i.e. the GAL4 strains used might differ from these points of view).

We thank the reviewer for the thoughtful and constructive comment. We refer the reviewer to our responses to the four main points above.

First, we point out that we have now shown reproducibility of the point of inflection (Figure 2C and Figure 4C).

We have conducted the experiments on aged flies and demonstrate that S-curve fit is not possible, either due to an already-decayed amplitude of oscillation, or a decay in amplitude so rapid that an Scurve fit is not possible due to the formation of infradian rhythms (i.e., there are too few peaks and troughs to fit an S-curve to).

With regard to the point about genetic backgrounds, we also demonstrate that the point of inflection in DvPdf-Gal4 driven LABL flies that falls to ~5 days is absent in aged flies, suggesting that the presence of this point of inflection is dependent on age and not exclusively on the genetic background of the fly expressing the Gal4 driver in this experiment. However, we agree that genetic background can certainly have an influence on our data when comparing between flies under the same conditions, and we therefore highlight in our manuscript that we cannot rule out the possibility that the contributions of genetic background influence where the point of inflection falls in the statements:

“Although we have not determined the extent of influence the genetic background the GAL4 drivers have on the A50, in all cases elimination of functional PDFR (all other genetic background being the same) shifted the A50 earlier in constant dark conditions.” (p. 15)

“A third possibility is that the differences of Gal4 drivers, or subtle genetic differences in fly backgrounds are causing the formation of 60-hour infradian oscillations.” (p. 23).

– One of my biggest concerns revolves around the experiments and the conclusions on PDF signaling. First, I did not fully understand why the authors chose to perform their experiments in pdfr-/- (han-/-) background instead of pdf-/-, especially knowing that PDFR might be responsive also to other neuropeptides (i.e. DH31) (an aspect correctly reported by the authors). Yet, most importantly, I am not convinced their conclusions about the negligible effects of PDF signaling on peripheral clocks are supported by the experimental plan used. To my knowledge, pdfr has never been shown as expressed in gut, fat bodies, and muscle (flybase reports expression in head and crop). More in general, PDF signaling has not been widely considered to synchronize/coordinate peripheral clocks (outside the head and prothoracic gland). This is a plausible assumption, but without data confirming the expression of pdfr in the tissues included in the analyses, no conclusion based on PDF signaling effects in these peripheral tissues can be drawn (i.e. "This result suggests that the presumed dominance of the brain clock in regulating peripheral clocks needs to be re-examined"). Thus, I encourage the authors to present evidence for such expression pattern (i.e. using immunohistochemistry), if this has not already been reported in the literature.

The reviewer makes a good point regarding our assumptions of PDFR expression in organs and with how PDFR and peripheral clocks interact. We have clarified our language in the text.

Since our initial submission, Li et al., have published a paper that describes all detectable transcripts that are expressed in various organs in flies. They report the expression of *pdfr* in a number of peripheral organs. We have cited this paper and incorporated it into our results (p. 19). We now refer to this paper as evidence that some peripheral organs express pdfr mRNA, but we underscore that mRNA expression does not reflect how much PDFR protein (or whether an effective amount of PDFR) is expressed. (pp. 20-21)

Our use of han5304 flies instead of pdf01 flies was for genetic reasons. Our supplemental table illustrates the number of transgenes involved in the genotypes of some of the flies that we used in our system. Using pdf01 would have required us to generate a large number of recombinant lines, and since our intention with this manuscript is to show that LABL is a functional reporter that generates reproducible luminescence signal to describe distinct clocks, we reasoned that the han5304 mutant would be an acceptable and a genetically convenient mutant candidate.

We agree that PDFR protein has not been shown to be expressed in the gut, fat body and muscle and we had not intended to imply that it had been. To avoid confusion by other readers, we changed the language in this section of the results to reflect that PDFR has not been shown to be expressed in these organs to make clear to the readers that we did not intend to suggest that we thought these organs expressed PDFR. We do however suggest that PDFR must play some sort of role in regulating peripheral clocks, not least because of transcripts reported in a number of peripheral organs in flies (Li et al., 2022), but also because luminescence oscillations from LABL flies that have been activated using organ-specific drivers reveal changed oscillation characteristics (relative to wild type).

In the abstract, the sentence that the reviewer highlighted “This result suggests that the presumed dominance of the brain clock in regulating peripheral clocks needs to be re-examined” was changed to simply state “we found that, while peripheral clocks in non-neuronal tissues were less stable after the loss of PDF signaling, they continued to oscillate.” Other parts of this region of the abstract were changed further to reflect the reworked Results section.

We changed the subtitle in the Results section from “Peripheral clocks exhibit distinct oscillations and variable PDF-dependence.” to “Peripheral clocks exhibit distinct oscillations and partial PDF-dependence.”.

Although we are excited by the prospect of showing PDFR expression in these organs as the reviewer suggests, given the difficulty in raising antibodies against PDFR, and given that flies express three alternatively spliced versions of PDFR, which complicates this endeavour (Paul Taghert, personal communication, SRBR 2022), and most importantly, since Paul Taghert’s group at Washington University is focused on this endeavour, we will not look for PDFR expression in peripheral organs.

We hope that the reviewer will find our changes satisfactory. We would like to underscore that the Taghert group themselves had difficulty in identifying expression patterns of PDFR in the brain for years and had to infer PDFR expression through the use of mutants and change in cell responses. We therefore think it is reasonable for us to demonstrate that there is an effect on clocks in peripheral tissue in the absence of functional PDFR, and emphasize that we do not know if this effect is direct, since we cannot demonstrate that PDFR is expressed in these tissues.

– My second major concern is about the lack of investigation and explanation of the ~60 h oscillatory dynamics identified. I understand that describing the ~60 h infradian oscillations goes beyond the scope of the study for the authors, yet, this represents such an atypical rhythm that I think the authors should provide a biological explanation about it, at least at the hypothesis/theoretical level. Moreover, I suggest discussing the findings concerning infradian rhythms unveiled in constant conditions in the context of the existing literature on ultradian and infradian rhythms unmasked by DD, LL, etc… On the other hand, graphs suggest infradian oscillations are actually very broad compared to the circadian ones (which are precisely around 24 hours) and range from ~48 h to ~72 h. Thus, I think it is not appropriate to talk about ~60 h rhythms. Moreover, this broad rhythmicity identified at the luminescence level might be of little to no biological meaning, being simply asynchronous, noisy, and stochastic oscillations. Maybe the authors could test whether this rhythmicity is present at the locomotor activity level as well. I think this would demonstrate that this infradian rhythmicity has a real link with *Drosophila* biology. However, also if this was the case, could we really consider it meaningful rhythmicity in constant conditions?

We thank the reviewer for the insightful comments and the suggestion of looking for infradian oscillations in locomotor behaviour – this was a very good idea that we had not considered. We believe that we have addressed the reviewer’s points in the manuscript, and conducted additional experiments, as detailed above in the main points. We refer the reviewer to our response under the four major points, above.

Ultimately, we agree about the infradian oscillations that we see, which is why in the original iteration of this manuscript, we “considered them arrhythmic from the circadian point of view”. However, we admit that these data are nonetheless interesting, which is why, with the encouragement of the reviews, we have expanded on them. We respectfully disagree that these signals can be noise. We see the long oscillations in the luminescence measurements visually, and when we quantify them, they are consistently around the ~60-hour period mark. We do not see how stochastic and noisy and synchronous oscillations can collect in a range reproducibly, only in han5304 mutants and not in tim01 mutants. Further to this, we do not see the han5304 pattern in tim01 flies (Figure 3B) nor in per0 flies (data not shown), suggesting that there is indeed a biological relevance to our observation. We also point out that wild type period fits range from 23 hour to 24.5 hours, while the majority of period fits for infradian rhythms fall between 55 and 63 hours (Figure 3C, inset, grey area).

We would also like to mention that Nave et al., (J Neuro, 2021) has recently shown that DN1s can oscillate with a very long period in constant light conditions in a cryb genetic background. We highlight this paper in the Discussion with the statement “We also point to seemingly long-period oscillations observed in the DN1s, in constant light conditions in a *cry*^b^ genetic background (Nave et al., 2021). Although this mechanism is likely different, it is another example of longperiod oscillations in the DN1s, which we observe in a han5304 background in DN1s after 6 days of constant darkness (Figure 6D).”, p 24.

Having argued this point though, we are grateful that the reviewers asked us to dig deeper into how these rhythms might occur. We hope that the reviewer feels that our explanations above of how an infradian rhythm may arise is satisfactory, particularly the possibility of strengthened coupling causing these longer rhythms – a hypothesis we are very excited about testing in the future.

Reviewer #2 (Recommendations for the authors):This manuscript describes a novel intersectional approach (LABL) that allows to express a per-luciferase driver in specific cell types or tissues in *Drosophila*. This is definitely a welcome addition to the arsenal of tools to study *Drosophila* circadian rhythms. Overall , the authors make a strong case that their approach works well. They are even able to record from a few brain neurons. The authors then use their novel method to test the impact of loss of PDF-receptor (PDFR) signaling on brain and peripheral oscillators. There, I have difficulties with the analysis and interpretation of the data, particularly when it comes to defining some form of hierarchy between circadian pacemaker neurons and peripheral tissues. There is already considerable literature on this, but the authors need to better link their results to current knowledge, and draw clear conclusions on whether their results indeed reveal something new. Results are not always coherent, and I am concerned that one of the main measurements used in these studies, called "amplitude stability constant", is not a reliable method to study the decay in rhythmicity. In summary, this paper brings a novel technology to the study of circadian rhythms in *Drosophila*, but its use to study PDFR signaling across fruit flies seems to provide rather limited new insight into the hierarchy between brain and body clocks, at least as currently presented.Specific comments:1) In the abstract, the authors say their results challenge the notion that brain clocks are dominant over peripheral clocks. However, it has been quite clear for a long time that peripheral clocks in *Drosophila* function independently. Most have their own photoreceptors, and do not depend on the brain for their oscillations under LD and for a few days at least in DD. There are a few exceptions though. An interesting one is the oenocytes, which receive PDF signaling (Krupp 2013). However, even there, rhythm amplitude is not affected by loss of PDF signaling, only phase. The Prothoracic Gland is however dependent on sLNv signaling for its rhythms, but this is through sNPF (Selcho 2017). The authors need to carefully survey relevant literature and place their results in context. It would be really interesting to try an oenocyte-GAL4 line and test whether LABL can pick up changes in phase of rhythms in the absence of PDFR in oenocytes. This would be a nice validation of this new approach.

The reviewer makes a very good point, and we have corrected our language surrounding this issue. The abstract was changed to more correctly reflect our conclusions. We have also made changes regarding our discussion involving PDF, as outlined above, under the four major points.

We agree with the reviewer that the oenocyte data would be very interesting. Since we are focusing this manuscript as a method and to turn our revisions around within a reasonable amount of time, we will conduct these (and other) experiments for reporting in a future manuscript, or in collaboration with interested labs.

2) In the introduction, the authors mention the lack of tissue-specific methods to monitor circadian rhythms in real-time. In flies, such methods are indeed lacking, but not in mammals. Intersectional approaches have been developed by the Schibler (Sinturel 2021) and Weaver (Smith 2022) labs, and those should be mentioned in the introduction. Sinturel should also definitely be mentioned in the discussion as their results are very directly relevant to the present study.

We have included the suggested references in our manuscript. We thank the reviewer particularly for the recommendation of the paper from the Schibler lab. We have modified the Results section, pp 19-20 to include the paper by Sinturel et al., 2021. We have included both Sinturel et al., and Smith et al., in our introduction (p 4)

3) The S-curve fitting of peak and through to determine rhythm decay does not appear reliable to me. This approach is going to be very sensitive to experimental noise, given how flat these curves are. Several curves do not look like an S (see Figures 5 and 6). How reproducible were these decay measurements across experiments? There are no statistics on Figure 5B and 6C. I am also not sure that the term "amplitude stability constant" is appropriate. This is not a constant.

When conducting experiments for the old and young flies, we had an opportunity to test the reproducibility of the point of inflection measure and S-curve measure for TUG-driven LABL flies (Figure 2C and Figure 4C). We note that these graphs are nearly identical, speaking to their reproducibility.

We went a bit further and attempted to measure the S-curve fit in individual experiments. This had mixed results. The robustness of the data is in the numbers, and S-curve fits into single, or two data points is very unreliable and non-informative. We therefore avoided such fits. Importantly, when we simultaneously run internal controls with experimental flies, we consistently reproduce trends: For example, an advanced point of inflection when flies are lacking a PDFR is always advanced.

Because of the suggestion of the reviewer, we have changed the “amplitude stability constant” to a more reflective A50, for Amplitude and 50%, inspired by IC50. This point represents the halfway point between the maximum and minimum values of the S-curve, where the point of inflection lies. We have made the changes to the manuscript accordingly.

4) I am not convinced that the infra-dian oscillations are absent in tim0 flies, just that they are less robust or synchronized than in pdfr-han. These oscillations might be the result of a weak underlying behavioral rhythm such as feeding, which could lead to low amplitude cycles of luciferin in the body. The weak inverted rhythms that appear after a few days in the esg-GAL4 experiment on Figure 6 could also be due to rhythmic ingestion and could explain the inversion of oscillations.

We feel we have addressed the reviewers concerns above, under the main points.

However, we thank the reviewer for their keen eye in noticing the inverted rhythms. We noted this, and speculated in-house that this may be similar to inverted rhythms observed by Yoshii et al., 2009 in pdf01 flies, but felt that we should follow up this point in a follow up manuscript. We have since decided to highlight this point and cite the Yoshii paper. (p. 21)

We have also added statistical analysis for graphs in insets of Figures 3B and 3C to demonstrate statistically that the infradian rhythms observed in data from han5304 flies are distinct from the fits in data from tim01 flies.

5) In the image for luciferase signal in the LNvs on Figure 4C, the signal seems to be too dorsally located to be from the LNvs. Even in the DvPDF image, the localization seems too dorsal. Would there be a luciferase antibody that could be used, to directly visualize LUC expression by ICC? The G-trace images look good, but they do not show directly luciferase expression.

We tried very hard to do this experiment over the span of a year. We tested four commercially available luciferase antibodies from abcam, novusbio, sigma and thermo-fisher. These did not work (we tested these antibodies in non-LABL controls, such as *plo* flies and UAS-Luc flies as well, and these too did not work). We were careful to collect flies for ICC imaging at the peak of measured luminescence to maximize luciferase expression. We then spent 6 months to try to produce our own Luciferase antibody, twice. Unfortunately, this also did not yield reliable ICC visualization. This is when we turned to G-trace to monitor FLP activity in target neurons and to luminescence image brains, as a compromise.

6) On page 11, the authors seem to conclude that body clocks need PDFR signaling, which contradicts one of the abstract's main statements. Then, the results in figure 6 seem rather different. Overall, loss of PDFR signaling does not seem to have much effect on figure 6 in peripheral clocks. How are those results reconciled?

We agree with the reviewer that our writing was confusing. We have cleaned up the language in this section, as described above.

[Editors’ note: further revisions were suggested prior to acceptance, as described below.]

Reviewer #1 (Recommendations for the authors):I personally consider the manuscript entitled "Real time, in vivo measurement of neuronal and peripheral clocks in *Drosophila melanogaster*" significantly improved overall. The authors conducted many suggested experiments and strengthened at the same time the weak points in the interpretation of generated data, phrasing, and literature gaps.The authors tackled with appreciated effort all the 4 major points of concern raised by the reviewers. Particularly, the authors effectively answered (a) my criticism concerning the misleading statements on the applicability/potential of the methodology developed in disease contexts, and (b) all the points I raised in my initial review.Concerning my criticism of the inflection point, the authors definitely strengthened the reproducibility of their analysis but failed to provide convincing evidence on clock stability in aged flies and the effect of the genetic background. However, I appreciated the dedication and effort in the experiments I suggested. I understand the failed aged flies` experiment is caused by insurmountable technical aspects, and I am satisfied with the transparency they used in openly addressing the genetic background issue I raised.

We thank the reviewer for this feedback. We also hope that our response to Reviewer 2’s item #2 reflects our acknowledgment of the point made by the Reviewer regarding genetic background and our measurement of point of inflection (A50).

Similarly, I positively value how the authors addressed my concerns regarding their interpretation of the role of PDF signaling. I understand the "genetically-convenience" reasons (considering their major goal), and the motivations for why they did not directly tackle PDFR expression, and I appreciate the implemented changes in the phrasing of some potentially misleading sentences.

We thank the reviewer for this feedback and in helping us clarify our statements.

Concerning the ~60 h rhythm concerns, although the experiments did not provide positive evidence, I again appreciate the authors´ investment in experimentally addressing these criticisms. At the same time, from a reader standpoint, the manuscript transparency is significantly improved, thanks to the inclusion of these data as supplementary information, and the theoretical explanations provided in the text. To be completely honest with the authors, I still feel the genetic background might have a bigger role than we think, yet, I do think the oscillator coupling argument is very valid and definitely the most plausible hypothesis, and I share their enthusiasm about future studies in this direction.

We once again thank the reviewer for guiding us to strengthen our manuscript and their careful insights. We have taken the suggestions to heart.

Finally, I agree with the editors about moving the manuscript to the "Tools and Resource" section of eLife. As much as the authors conducted interesting/impactful research and careful analyses, I still feel the advancement provided in the manuscript from a biological standpoint is still surpassed by the potential of the methodology the authors developed.All considered I endorse the publication of this manuscript on eLife.

We thank the reviewer for their positive and constructive feedback in helping us strengthen our manuscript.

Reviewer #2 (Recommendations for the authors):The authors have improved their manuscript, though some issues remain. I will first go through the four main issues that were brought up in the first round of reviews.1) Validation of the approach: This has certainly been improved with the use of clock mutants and old flies. LABL should prove to be a potent new tool for studying circadian rhythms in *Drosophila*.

We thank the reviewer for this feedback.

2) I still have concerns with the reliability of the inflection point. It is reassuring that the authors observed a similar A50 in two independent sets of experiments done with wild-type flies. However, they acknowledge that there is important variability between individual experiments. I am not sure that inflection points will prove reliable beyond robustly rhythmic genotypes. For example, in figure 7, it does not appear that perL can be fitted with an S shape. Also, by not being able to use individual flies or even individual experiments, there is no way to test differences between genotypes with statistical analysis. Some of the differences that were observed might not hold in future experiments. The authors should acknowledge those important limitations.

We agree with the reviewer’s assessment, and we hope that the additional modifications also satisfy any remaining concern from Reviewer 1.

On page 11, we added the sentence, “However we note that since we do not determine A50 for individual flies, this measure does not readily permit statistical analysis for comparison across genotypes.” to underscore the statistical limitation of determining A50 on a collection of flies rather than individual flies.

On page 11, we changed the sentence “Thus, we conclude that an A50 could not be determined for aged flies because the amplitude in their endogenous clock oscillations had already decayed to their minimal stable levels.” to “Thus, we conclude that an A50 could not be determined for less stable clock oscillations in aged flies because the amplitude in their endogenous clock oscillations had already decayed to their minimal stable levels.” We believe this underscores the caveat for A50 analysis.

On page 15, in the section under perL analysis, we added the sentence “The apparent noisiness of the oscillation prevented us from determining an A50, which suggests that this parameter is limited to more stable oscillations.” To underscore the limitation for A50 analysis.

3) Discussion of previous results on PDF and PDFR signaling has been improved, but I noticed issues with referencing that will be detailed below.

We hope that we have addressed all concerns below.

4) The lengthy response of the authors to the questions about the infradian rhythms is at times difficult to follow. In particular, I do not understand why they exclude some form of metabolic/feeding rhythm. These rhythms might simply be revealed when circadian control is lost or severely impaired. They might be more clustered around 60 hrs in han mutant flies compared to tim0 , because tim0 flies have no circadian clock at all.

We apologize for not making our argument clearer. We suggest that metabolic/feeding rhythm is an unlikely explanation for the observed 60-hour rhythms in han5304 flies.

Reason 1: LABL activated in pdf+ neurons reveal luminescence oscillations with a period in the vicinity of 24 hours. LABL activated in R18H11+ neurons reveal a decay to a ~60-hour rhythm on day 5 of constant darkness. Other drivers (e.g., clk9M, Mai179) exhibit different days at which the 24-hour rhythm falls into an ~60-hour rhythm. Since the main genetic difference between them is the driver that they carry, we expect each type of fly to exhibit eating patterns similar to each other. We therefore conclude that ~24-hour oscillations and ~60-hour oscillations from flies that are otherwise genetically identical are unlikely to be caused by a change in eating behaviour.

Reason 2: Another explanation for a 60-hour luminescence oscillation may be that luciferin is absorbed into the different tissues differently, rather than being exclusively correlated with luciferase expression. However, this would require that differences in luciferin absorption in different tissues must exist, that it is also clock dependent (as tim0 does not show 60-hour oscillations as the Reviewer correctly points out), and that the 24-hour transcription oscillation and rhythmic luciferin absorption conspire to reveal a 60-hour rhythm.

While we cannot rule these possibilities out, we suggest that changes to coupling strength of different 24-hour transcription oscillation offers a simpler and more likely explanation.

We hope that this clarifies our thinking. We do agree that our arguments do not rule out contribution from metabolism, but we hope that we are able to express that we suspect that it is not a major contributing factor.

Reviewer #3 (Recommendations for the authors):The authors have done a good job in responding to my previous critiques. They have added new experiments, new analyses, new model simulations, and new text. The changes to figures have made them clearer. While it remains unclear to this reviewer how the 60-hour LABL rhythms arise or what they mean, the discussion of possible explanations is thoughtful and the source/meaning of these infradian rhythms remains outside the scope of this paper. The genetic tools generated in this paper have been well validated and characterized and will be appropriately published in the Tools and Methodology section of eLife. Without a doubt, these tools will be highly useful for *Drosophila* researchers in circadian biology and this manuscript is likely to be well cited.

We thank the Reviewer for their positive and constructive feedback. We believe their feedback has significantly improved this manuscript.